# Parameters estimation of stochastic finite fault ground motion simulation method and its application in North China

Xiaohui Jia [1]*, Zihan Feng[1], Xiaoshan Wang[2], Aiwen Liu[3]

**1** Hebei GEO University, Hebei Technology Innovation Center for Intelligent Development and Control of Underground Built Environment, Shijiazhuang, China, **2** Hebei Earthquake Agency, Shijiazhuang, China, **3** Institute of Geophysics, China Earthquake Administration, Beijing, China

* jiaxiao_hui@126.com

## Abstract

The stochastic finite-fault method based on dynamic corner frequency has been widely applied to simulate high-frequency ground motion of near-fault field. The model input parameters include source term, path term and site condition, many of which exhibit strong regional properties and are particularly sensitive to the site. Based on strong-motion stations drilling data and recordings within North China, some region-specific key parameters including local site amplification and high-frequency decay factor kappa are examined and analyzed. The local amplification function over versus frequency for class II and III site are computed using the quarter-wavelength approximation. The kappa in North China Plain and Mountain regions are calculated by employing the spectral decay method, respectively. In addition, the calibration of stress drop is achieved by adopting the trial-and-error method, which can be also applicable to the determination of other uncertain model parameters. After input parameters are determined and model bias is evaluated, we applied the region-specific parameters in North China to simulate and analyze time history, peak ground acceleration, Fourier amplitude spectrum, acceleration response spectrum, ShakeMap of PGA for the Dezhou earthquake. Overall, the simulated results coincide well with observed recordings. The validation of region-specific parameters demonstrates that they could be applied to the synthetic high-frequency ground motions in North China.

## Introduction

Earthquake is the main natural disaster which may induce building damages and loss of life. Since the 1970s, a series of destructive earthquakes around the world have occurred. For some examples, the 1976 Tangshan $M_w$ 7.5 earthquake in North China caused nearly 250,000 deaths and most of building collapses [1]. In 1994, the

**Data availability statement:** The finite-fault ground motion simulation program code can be downloaded from website of Engineering Seismology Toolbox (https://www.seismotool-box.ca/EXSIM12.html). Some seismic parameters are available at website of USGS (https://earthquake.usgs.gov/earthquakes/eventpage/us6000ky5l/executive). Strong motion recordings for this study are provided by China Strong Motion Network Centre at Institute of Engineering Mechanics, China Earthquake Administration.

**Funding:** Doctoral Research Initiation Foundation of Hebei GEO University under Grant Nos. 2024074.

**Competing interests:** The authors declare no competing interests.

Northridge $M_w$ 6.7 earthquake in Los Angeles led to 58 deaths and more than 600 people injured [2]. In 1999, the Taiwan Chi-Chi $M_w$ 7.6 earthquake caused over 2,400 deaths and 100,000 house collapses [2]. In 2008, the Wenchuan $M_w$ 7.9 earthquake in Southwest China shook half of China and many countries in Asia, resulted in nearly 90,000 fatalities and incalculable economic losses [3]. The post-earthquake investigation from these past earthquakes have shown that strong ground motion is the crucial impact factor, which can cause building damage, loss of life and ground failure, including landslide and liquefaction. In other words, ground motion time histories are essential input parameters for seismic design and dynamic analysis of building structure. In the regions where there are few strong motion records, it is very useful to simulate acceleration time histories as the input. In addition, if peak ground acceleration (PGA) field near seismic fault can be estimated before a possible earthquake in the future, seismic fortification and emergency measures can be formulated to avoid casualties and reduce earthquake loss. If PGA field can be obtained in the shortest time after earthquakes occur, we can assess earthquake intensity and take effective measures to rescue and recovery in post-earthquake promptly. Consequently, the prediction of near-field strong ground motion is always a critical research topic to be studied in the field of earthquake engineering.

The stochastic simulation method is a simple and powerful tool, which combines seismology theory and some empiricism in predicting ground motion, and which has obvious advantage of requiring little information on source slip distribution and crustal structure. Therefore, the stochastic method not only performs good for past earthquakes, but is a valuable simulation tool for predicting future earthquake events with unknown modeling input parameters. So far, it has been widely used in different seismic tectonic environments, including Eastern North America [4,5], California [6], Greece [7], Russia [8], Iran [9], Japan [10], China [11–14], India [15]. In our opinion, the successful application of stochastic method is the driving force for us to continue working on its development.

Stochastic method of simulation (SMSIM) was first proposed by Boore [16], which treated the source as a point-source. There were significant limitations in SMSIM, and it could not be applied to large earthquakes and near-field of small-medium earthquakes. To overcome these drawbacks, a finite-source simulation (FINSIM) was put forward, which subdivided the fault plane into several small subfaults [16]. Therefore, the final synthetic acceleration time history was composed of the acceleration produced by each subfault. However, the subfault size had a significant effect on the simulation results and conservation of seismic moment. Consequently, a finite-fault modeling based on dynamic corner frequency (EXSIM) was proposed and improved continuously by researchers all over the world [17]. Since then, the EXSIM was widely applied to simulate ground motion.

North China, where witnessed several moderate and large earthquakes with serious casualties, has a relative active seismicity tectonic environment. The focus of this study is based on 15 earthquakes and dozens of stations in North China, listed in Table 1, Table 2 and depicted in Fig 1. In this article, we applied the stochastic finite-fault method based on dynamic corner frequency proposed by Motazedian and

**Table 1. Information on the selected earthquakes used in this study.**

| Event No. | Date (yyyy-mm-dd) | Location | Lat (°N) | Lon (°E) | Magnitude ($M_s$ or $M$) | Depth (km) | Reference |
|---|---|---|---|---|---|---|---|
| 1 | 2010-03-06 | Luanxian | 39.700 | 118.300 | $M_s$ 4.3 | 10 | CENC |
| 2 | 2010-04-09 | Fengnan | 39.509 | 118.110 | $M_s$ 4.1 | 13 | CENC |
| 3 | 2012-05-28 | Tangshan | 39.709 | 118.470 | $M_s$ 4.7 | 22 | CENC |
| 4 | 2012-06-18 | Baodi | 39.610 | 117.559 | $M_s$ 4.0 | 5 | CENC |
| 5 | 2014-09-06 | Zhuolu | 40.270 | 115.419 | $M_s$ 4.3 | 20 | CENC |
| 6 | 2015-09-14 | Changli | 39.700 | 118.800 | $M_s$ 4.2 | 14 | CENC |
| 7 | 2015-11-28 | Fengnan | 39.299 | 117.900 | $M_s$ 3.4 | 5 | CENC |
| 8 | 2016-06-23 | Shangyi | 40.950 | 114.199 | $M_s$ 4.0 | 14 | CENC |
| 9 | 2016-09-10 | Tangshan | 39.689 | 118.319 | $M_s$ 4.0 | 10 | CENC |
| 10 | 2018-02-12 | Yongqing | 39.369 | 116.669 | $M_s$ 4.3 | 20 | CENC |
| 11 | 2019-12-03 | Huaian | 40.450 | 114.550 | $M_s$ 3.4 | 13 | CENC |
| 12 | 2019-12-05 | Fengnan | 39.310 | 118.040 | $M_s$ 4.5 | 10 | CENC |
| 13 | 2019-12-23 | Jizhou | 39.849 | 117.330 | $M_s$ 3.3 | 10 | CENC |
| 14 | 2020-07-12 | Guye | 39.779 | 118.440 | $M$ 4.8 | 10 | CENC,USGS |
| 15 | 2023-08-06 | Dezhou | 37.159 | 116.339 | $M$ 5.5 | 18 | CENC,USGS |

*The magnitude $M$ used in this article is the moment magnitude based on the results released by U.S. Geological Survey (USGS) and $M_s$ is the surface wave magnitude released by China Earthquake Networks Center (CENC), respectively.

Atkinson to estimate the region-specific seismic parameters in North China [17]. Based on the borehole data and ground motion recordings, some region-specific key parameters including local site amplification, high frequency attenuation kappa and stress drop were calculated and analyzed, and some parameters including geometric spreading function and anelastic attenuation (Q-value), were adopted from previous studies or empirical relationships. After the input parameters were determined, we applied region-specific key parameters in North China to simulate and analyze time histories, peak ground acceleration, Fourier amplitude spectrum (FSA), 5%-damped pseudo-acceleration response spectrum (PSA), ShakeMap of PGA for the Dezhou earthquake.

### Strong motion database

With the continued construction of strong motion stations over the last fifteen years, the China Earthquake Administration operates more than 100 stations equipped with three-component strong motion seismograph in North China. Since its preliminary inception in 2010, the ground motion database and station drilling data have been growing steadily because of the occurrence of earthquakes and increasing number of stations. So far, the China Strong Motion Network Center (CSMNC) has recorded at least 15 earthquake events with magnitude from 3.3 to 5.4 and released more than 1000 acceleration time series in study area.

In this article, acceleration time series for the 15 earthquake events that we compute and analyze are presented in Table 1, and strong motion stations used in this study are listed in Table 2, respectively. Fig 1 shows the epicenters of study earthquakes and the locations of strong motion stations in North China study area.

The sample rate of original acceleration time history was 200 Hz, and the preceding 20 s was prestored as the pre-event part. In the course of data processing, all recorded acceleration time histories were baseline-corrected to remove the mean and linear trend firstly. Then, a fourth-order Butterworth filter was used to band-pass-filtered in the frequency range of 0.02-20 Hz, which contained the main destructive seismic ground motion characteristics. Followed by band-pass-filtered, the shear-wave portion of whole time series were windowed before the zero-distance kappa factor ($\kappa_0$). Then, the Fourier amplitude spectrum (FAS) and the 5%-damped pseudo-acceleration response spectra (PSA) were computed.

**Table 2. Information on the strong motion stations used in this study.**

| Station Code | Lat (°N) | Lon (°E) | Site class (China) | Site class (NEHRP) | Sensor Type |
|---|---|---|---|---|---|
| LGZ | 39.44 | 115.36 | II | B | ETNA/ES-T |
| HMY | 41.10 | 116.89 | II | B | ETNA/ES-T |
| GLZ | 40.71 | 114.55 | II | B | ETNA/ES-T |
| JEG | 40.14 | 114.29 | II | B | ETNA/ES-T |
| ZNB | 41.00 | 115.57 | II | B | ETNA/ES-T |
| GXC | 41.43 | 115.79 | II | B | ETNA/ES-T |
| YAJ | 39.93 | 116.83 | III | C | ETNA/ES-T |
| WBH | 39.67 | 117.05 | III | C | ETNA/ES-T |
| GGZ | 39.61 | 116.58 | III | C | ETNA/ES-T |
| BZZ | 39.62 | 116.57 | III | C | ETNA/ES-T |
| BGZ | 39.35 | 116.66 | III | C | ETNA/ES-T |
| DCF | 39.49 | 115.79 | III | C | ETNA/ES-T |
| XHY | 40.10 | 114.80 | Soil | Soil | ETNA/ES-T |
| ZJP | 39.90 | 115.30 | Soil | Soil | ETNA/ES-T |
| FHL | 40.10 | 116.10 | Soil | Soil | MR2002/SLJ-100 |
| SSL | 40.20 | 116.30 | Soil | Soil | MR2002/SLJ-100 |
| SDT | 40.00 | 116.20 | Soil | Soil | MR2002/SLJ-100 |
| SDZ | 39.60 | 115.60 | Soil | Soil | MR2002/SLJ-100 |
| STJ | 40.00 | 115.50 | Soil | Soil | ETNA/ES-T |
| SWD | 40.00 | 116.00 | Soil | Soil | MR2002/SLJ-100 |
| WJY | 40.20 | 116.00 | Soil | Soil | MR2002/SLJ-100 |
| XYT | 40.40 | 115.80 | Soil | Soil | MR2002/SLJ-100 |
| YJS | 39.60 | 115.80 | Soil | Soil | ETNA/ES-T |
| BTS | 40.40 | 115.20 | Soil | Soil | ETNA/ES-T |
| DBX | 40.10 | 115.10 | Soil | Soil | ETNA/ES-T |
| HHY | 40.20 | 115.30 | Soil | Soil | ETNA/ES-T |
| LAS | 40.40 | 115.70 | Soil | Soil | ETNA/ES-T |
| DOL | 40.20 | 117.70 | Soil | Soil | ETNA/ES-T |
| HOQ | 39.80 | 117.70 | Soil | Soil | ETNA/ES-T |
| WLG | 39.70 | 117.80 | Soil | Soil | ETNA/ES-T |
| JZG | 39.40 | 118.10 | Soil | Soil | ETNA/ES-T |
| QFT | 35.80 | 115.00 | Soil | Soil | ETNA/ES-T |
| XXT | 35.00 | 113.90 | Soil | Soil | ETNA/ES-T |
| FXT | 35.80 | 115.50 | Soil | Soil | GSMA-24IP/SLJ-100 |
| YJT | 35.10 | 114.20 | Soil | Soil | ETNA/ES-T |
| LYT | 34.30 | 112.20 | Rock | Rock | GSMA-24IP/SLJ-100 |

Table 2 lists the detailed information about typical stations that we analyze in this study, which includes latitude, longitude, digital instruments and site classification according to the China Seismic Code GB50011−2010 [18], and the United States National Earthquake Hazards Reduction Program (NEHRP).

## Simulation method (EXSIM12)

In earthquake engineering, previous studies have shown that strong ground motion is a complex result of contributions from the seismic source rupture process, wave propagation path in crustal medium and site effect [13]. In finite-fault

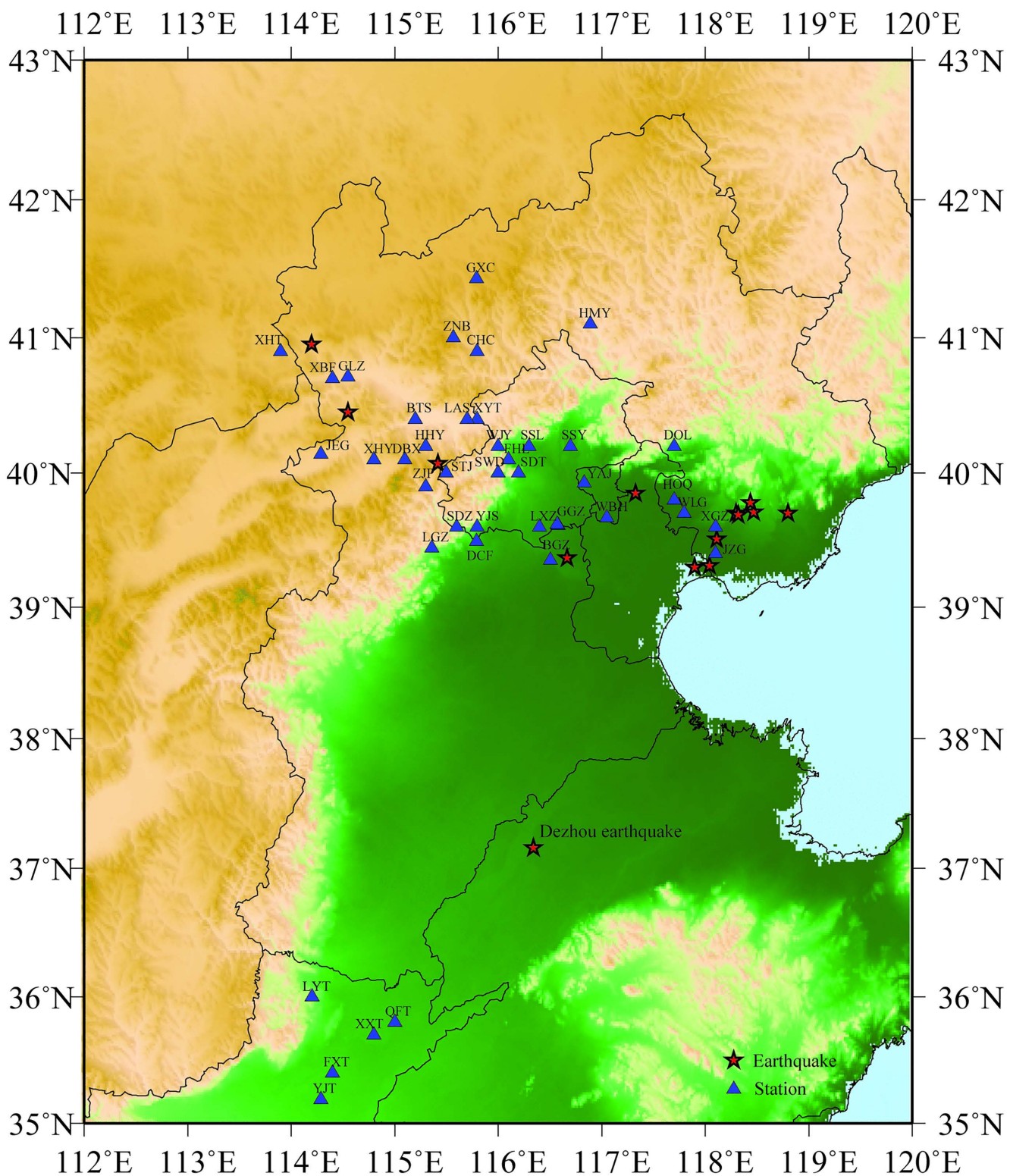

**Fig 1. Distribution of the study earthquakes and strong motion stations in North China study area.** The epicenters and selected stations are represented as pentagram and triangle, respectively. Fig 1 is plotted by the Generic Mapping Tools (GMT) version 5.4.4 (https://gmt-china.org/download/).

modeling, the rupture fault is divided into N subfaults, each of which represents a point seismic source [19]. The rupture starts from hypocenter and spreads radially to trigger nearby subfaults in turn. In stochastic finite-fault simulation method, the shear-wave Fourier amplitude spectrum $A_{ij}(f)$, which is the $ij$th subfault at an observation $R_{ij}$ away from source in frequency domain, can be defined by

$$A_{ij}(f) = C \cdot Source(M_{0ij}, f) \cdot Path(R_{ij}, f) \cdot Site(f) \tag{1}$$

In stochastic finite-fault modeling based on dynamic corner frequency introduced by Motazedian and Atkinson [17], the equation above can be further described as follows:

$$A_{ij}(f) = \left\{ CM_{oij}H_{ij}(2\pi f)^2 / \left[ 1 + (f/f_{oij})^2 \right] \right\} \cdot \left\{ G(R_{ij}) \cdot exp(-\pi f R_{ij}/Q\beta) \cdot exp(-\pi f \kappa_0) \cdot A(f) \right\} \tag{2}$$

In the Eq(1) and (2), $M_{0ij}$, $R_{ij}$, $H_{ij}$ and $f_{oij}$ are the $ij$th subfault seismic moment, distance from the observation point, scaling factor to conserve the high-frequency spectral level and dynamic corner frequency, respectively. The constant $C = R^{\theta\phi}FV/(4\pi\rho\beta^3)$, where $R^{\theta\varphi}$ denotes radiation pattern (shear-wave average value is 0.55), $F$ stands for free surface amplification (2.0), $V$ represents partition onto two horizontal components (0.71), and $\rho\beta$ are crustal density and shear-wave velocity. The dynamic corner frequency $f_{oij}$, as a function of time $t$, is expressed by $f_{oij} = 4.9 \times 10^6 \ \beta(\Delta\omega t\sigma/M_{oave})^{1/3} \cdot N(t)^{-1/3}$, where $\Delta\omega t\sigma$ indicates stress drop of earthquake [20], $N(t)$ stands for the cumulative number of ruptured subfaults at time $t$, $M_{oave} = M_0/N$ is the average seismic moment of $N = nl \times nw$ subfaults, in which $nw$ and $nl$ are the number of subfaults along the width and length of fault plane, respectively. The $ij$th subfault seismic moment $M_{oij}$ is controlled by the ratio of the $ij$th subfault area to the entire area of seismic fault. When the subfaults are not identical, $M_{oij}$ is given by $M_{oij} = M_0 S_{ij}/(\sum_{i=1}^{nl} \sum_{j=1}^{nw} S_{ij})$, where $M_0$ is the entire fault seismic moment and $S_{ij}$ is the relative slip of the $ij$th subfault.

In frequency domain, the path attenuation effect $Path(R_{ij}, f)$ is modeled by multiplication of distance-dependent geometrical spreading function $G(R_{ij})$ and anelastic path attenuation function $exp(-\pi f R_{ij}/Q\beta)$, where $Q$ is the quality factor. The anelastic attenuation frequency-dependent $Q(f)$ can be expressed in the form of $Q(f) = Q_0 f^{\eta}$, in which $f \ \eta$ are the frequency and the frequency decay parameter, respectively. The path effects depend on the different propagation paths between source and site, and dominate the attenuation characteristics of simulated ground motion.

The site effect generally includes the amplification factor $A(f)$ and high-frequency attenuation $D(f)$, which can be expressed in the form of $Site(f)A(f) \cdot D(f) = A(f) \cdot exp(-\pi f \kappa_0)$. In general, amplification factor $A(f)$ involves the upper crustal amplification and local site amplification. The term $exp(-\pi f \kappa_0)$ is a high-cut filter to model near surface high-frequency attenuation effect, which is commonly observed rapid Fourier spectrum decay of acceleration recordings at high frequencies [21].

Acceleration spectrum of the ijth subfault is Fourier transformed to time domain, and the ground motions of each subfault is calculated by the stochastic point-source method [16,20]. Then total ground motion acceleration $a(t)$ from the entire fault, can be obtained by summing up simulated motions from all subfaults with a proper time delay,

$$a(t) = \sum_{i=1}^{nl} \sum_{j=1}^{nw} a_{ij}(t + \Delta t_{ij}) \tag{3}$$

where $\Delta t_{ij}$ denotes relative delay time for radiated seismic wave from the $ij$th subfault to hypothetical observation station.

The above mentioned dynamic corner frequency modeling was proposed by Motazedian and Atkinson [17], and made further improvements on spectral amplitudes by Boore [16]. This study adopted the stochastic finite-fault open computer program in FORTRAN named EXSIM12, which was complied by Atkinson and Assatourians [22]. However, this method needs to build and input source parameters including stress drop and slip distribution, path parameters including quality

factor and geometric spreading behavior, and site parameters such as crustal amplification function, local site amplification and $\kappa$. In addition, most of these parameters are often different in specific region. Consequently, the more accurate region-specific input parameters we obtain, the more realistic acceleration time series with the period range of engineering interest we can reproduce.

## Parameters estimation in North China (EXSIM-NC)

In this study, some modifications or variation of EXSIM12 on site amplification and kappa in North China region, which is named EXSIM-NC, is proposed.

## Site amplification

In EXSIM, site amplification is a crucial parameter, which may amplify seismic waves in different frequency. As mentioned above, site amplification includes upper crustal and local site amplification. If strong motion station is located on bedrock, we can only use crustal amplification. If station is situated on soil site, such as site class depicted in Table 2, the local site amplification must also be taken into account. Currently, local site amplification can be calculated in four ways: (1) The Standard Spectral Ratio Approach, which demands to find a suitable bedrock site [23]. (2) The Horizontal to Vertical Spectral Ratio Method, which may underestimate the amplification [23]. (3) The Generalized Inversion Technique, which determines the shear wave velocity through an iterative process of the arrive time residual [13]. (4) The quarter-wavelength approximation [24], which is based on density and shear-wave velocity as functions of depth. Considering that we can obtain detailed site drilling profile in North China, the fourth method is chosen.

For a particular frequency, the amplification value is calculated by the square root of ratio between the seismic impedance of bedrock at the depth of source and the average seismic impedance from surface to a quarter wavelength, where seismic impedance is defined as shear wave velocity times density. The algorithm expression is the following:

$$A(f) = \sqrt{\frac{\rho_s \beta_s}{\overline{\rho_z}\,\overline{\beta_z}}}$$

(4)

where $\rho_s$, $\beta_s$ represent density and shear wave velocity at source, and $\overline{\rho}_z$, $\overline{\beta}_z$ represent average density and shear wave velocity at site, respectively.

In order to gain local site amplification over a wide range of frequencies, soil models with deeper velocities and densities versus depth beyond borehole are demanded. For class II site, borehole shear wave velocity and density are used from surface to 30 m, and the average shear wave velocity profile of shallow hard site in California is adopted from 30 m to the depth of 1500 m/s. For class III site, which is generally located on sedimentary plain, borehole measure data are still used from surface to 30 m, and the velocity profile below 30 m is established on the P-wave velocity structure of the crust in the North China Plain region [25]. Then, we use the quarter-wavelength approximation to compute and obtain the amplification function for each borehole listed in Table 2, as well as the mean amplification function values of the boreholes in class II and III, respectively.

In Fig 2(a) and 2(b), we compare the borehole shear wave velocity profile versus depth in North China with California. As can be seen, the velocity in this study is smaller than the result in California at the same depth, either in class B or class C site. Based on the velocity and density profiles, local site amplification function curves in class B and class C site are given in Fig 2(c) and 2(d), respectively. The result indicates that the trend of curves is consistent with fact that amplification factor increases gradually with the increase of frequency. In addition, corresponding estimated values in six fixed stations for each frequency are very close to each other in Fig 2(c) and 2(d), respectively. For class B site, at frequencies below 0.1 Hz, the amplification factor is 1 or slightly greater than 1; when frequency exceeds 30 Hz, the amplification factor is about 5 times. For class C site, when frequency is 30 Hz, the amplification factor reaches about 6 times. In Fig 2(e) and 2(f), the

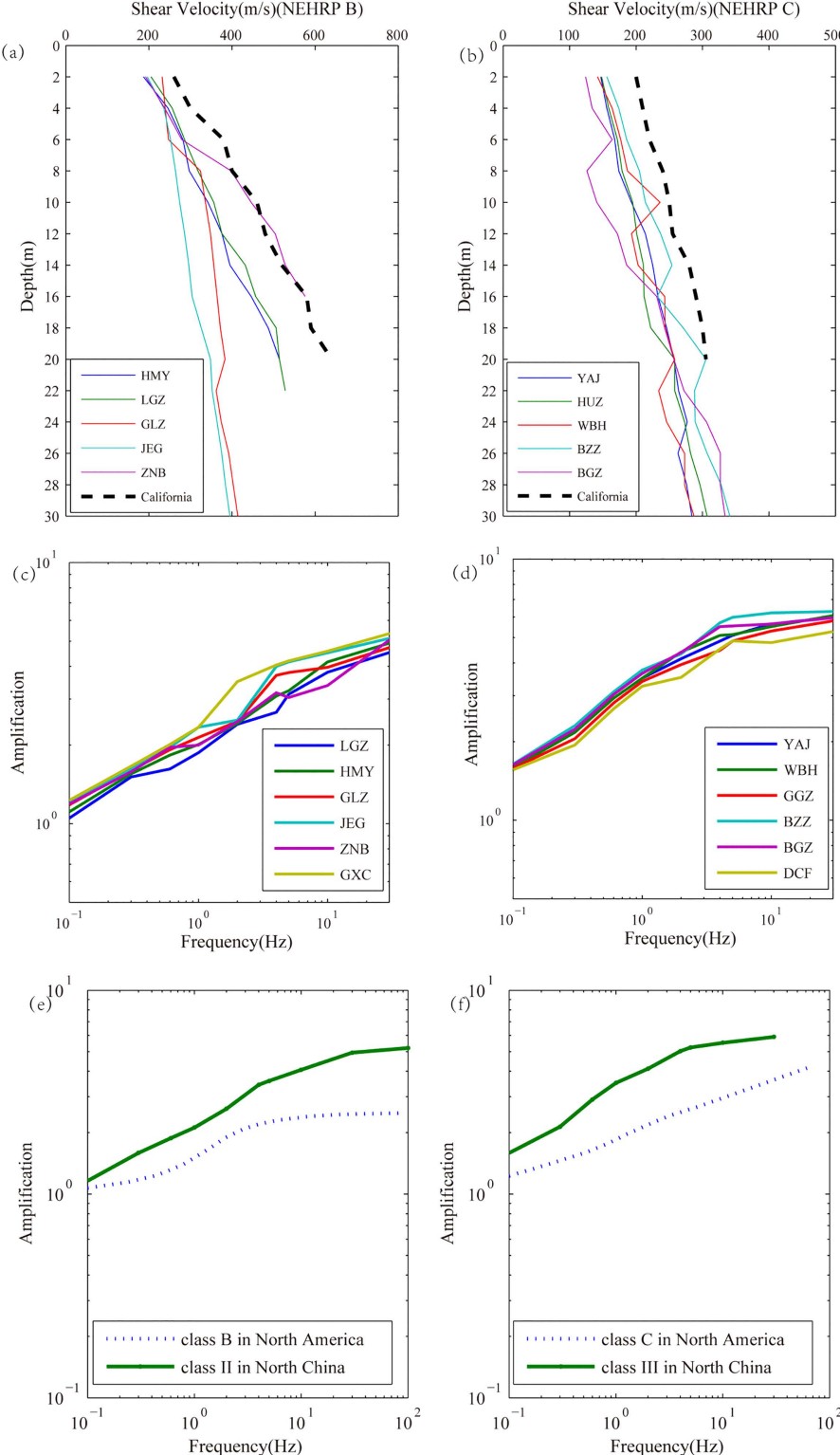

**Fig 2. Fig (a) and (b) show the comparison of shear wave velocity profile versus depth in North China and California.** Fig (c) and (d) display the local site amplification function curve for each borehole. Fig (e) and (f) show comparison of the mean amplification function curve between North China and North America.

comparison result presents that mean amplification function values estimated in North China are slightly higher than local site amplification function established in North America. The difference may be attributed to the different criteria for site classification and different characteristic of regional geology.

Furthermore, the amplification factor of station LGZ with average shear wave velocity of 394 m/s and sedimentary thickness of 22 m is the smallest in Fig 2(c), and amplification of station GXC with 265 m/s and 30 m is the largest, which indicates that the size of amplification factor has a great relationship with average shear wave velocity and sedimentary thickness. Overall, the lower average shear wave velocity and the thicker sedimentary layer are, the bigger amplification factor will be [24]. In addition, Fig 2(d) illustrates the same scenario.

**kappa ($\kappa$)**

High-frequency decay factor kappa ($\kappa$) is a critical parameter in ground motion simulation, which is mainly used to represent the high-frequency decay of Fourier amplitude spectrum above a specific frequency. The value of $\kappa$ can be determined by the high-frequency attenuation characteristics of fixed stations and propagation pathways, which could be expressed as follows:

$$\kappa = \kappa_0 + \text{m} \cdot R \tag{5}$$

where $\kappa_0$ is the zero-distance intercept of $\kappa$ factor, and m represents the coefficient dependent on $Q$ and $\beta$. In this article, we adopt the spectral decay method of Anderson and Hough [21], which is the most commonly used method for calculating $\kappa$. In general, the phenomenon that high-frequency decay factor $\kappa$ satisfied a simple linear approximation in FSA semi-logarithmic coordinates was found, and $\kappa$ could be obtained as follows equation:

$$\kappa = -\frac{\text{dln}A(f)}{\pi\text{d}f}=, \ f_1 \leq f \leq f_2 \tag{6}$$

where $f_1$ and $f_2$ denote the frequency at which high-frequency attenuation starts and ends, respectively.

An example of the $\kappa$ calculation process is shown in (Fig 3a-3b). After zero-line correction and filtering, both the noise (that is pre-event portions) and Shear-wave windows are extracted in Fig 3(a). Then, Fourier amplitude spectrum of these two portions are calculated by fast Fourier transform and plotted in a semi-logarithmic coordinate system. In the selection process of $f_1$ and $f_2$, the value of $f_1$ is generally taken near the high-frequency cut-off frequency, and the value of $f_2$ is taken as high-frequency attenuation termination point. The $f_1$ and $f_2$ of high-frequency attenuation are picked manually. To reduce the effect of local anomaly of Fourier spectrum on the result, the bandwidth between $f_1$ and $f_2$ cannot be less than 10 Hz. In this case, Fig 3(b) displays the $f_1$ is 8 Hz and $f_2$ $\kappa$ value of east-west (EW) component at station XHY is 0.0512 s.

In this study, the $\kappa$ for each individual recording is obtained based on the same procedures, and 81 three-component recordings on 27 stations are analyzed. The $\kappa$ calculated for the horizontal components of all recordings versus hypocentral distance are shown in (Fig 3c-3f). The distribution of $\kappa$ versus distance is linear fitted by the form $\kappa = \text{m} \cdot R + \kappa_0$. The best-fit coefficients for the north-south (NS) and east-west (EW) components in North China Plain region are $\kappa_0^{NS}$ =0.04762 s and $\kappa_0^{EW}$ =0.04987 s, which are plotted in Fig 3(c) and 3(d), respectively. In North China Mountain region, the relationship of NS and EW components can be expressed as $\kappa^{NS}$ =0.000130$R$+0.02205, and $\kappa^{EW}$ =0.000086$R$+0.02205, as shown in Fig 3(e) and 3(f). In addition, the R-Square values of the linear regression are reported in the figure.

To weaken the heterogeneity influence of propagation effect in horizontal components, the arithmetic mean kappa $\overline{\kappa_0}$ values of NS and EW components are further calculated, the result of which suggests a zero-distance $\overline{\kappa_0}$ of 0.049 s in North China Plain region and $\overline{\kappa_0}$ of 0.020 s in North China Mountain region, respectively. This result implies that the $\overline{\kappa_0}$ value has strong regional characteristics and is severely affected by the local site condition. More specifically, as the

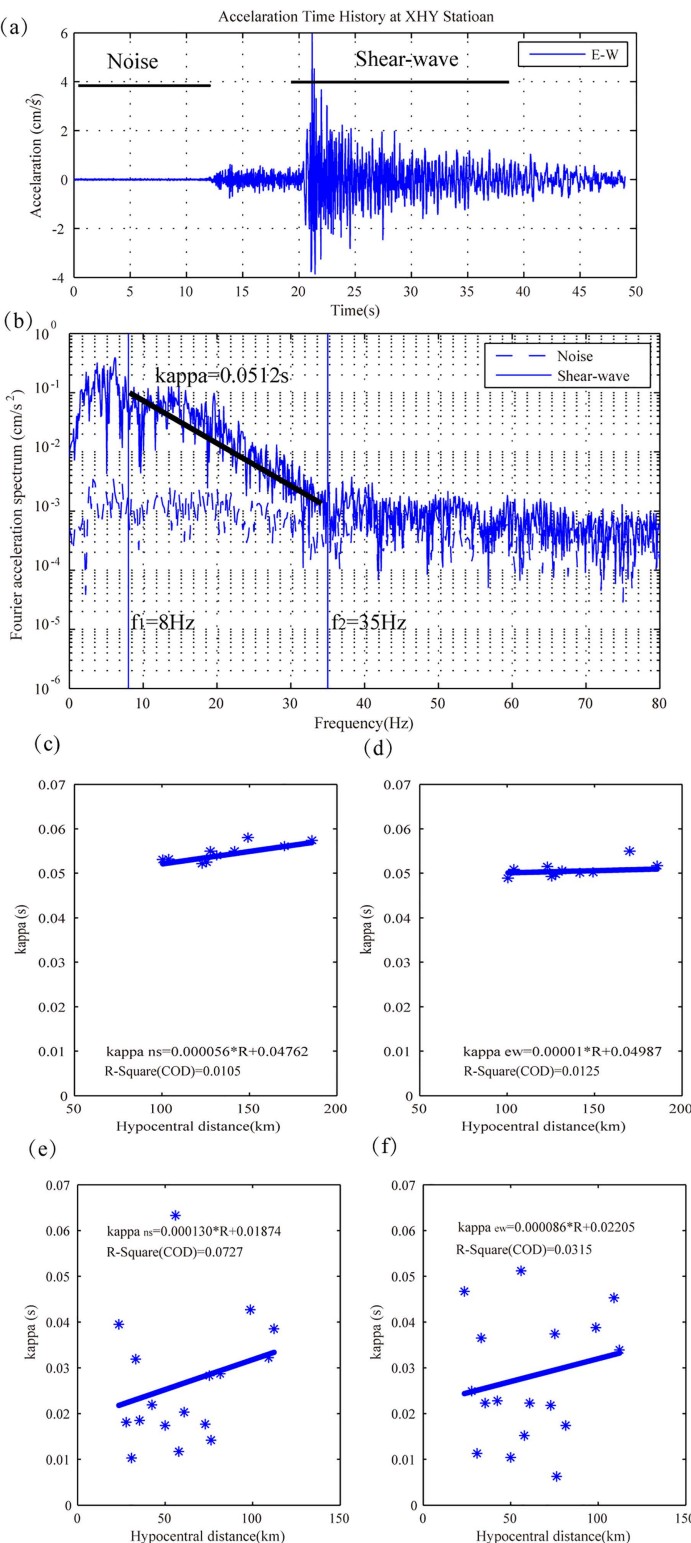

**Fig 3. Estimation of kappa value and distance dependence of horizontal kappa.** Fig (a) and (b) show an example of detail calculation process of κ value at station XHY. Fig (c-d) and (e-f) display the distribution of κ vversus hypocentral distance for horizontal components in the North China Plain region and Mountain region, respectively.

site condition changes from soft soil in sedimentary plain to hard rock in mountain, the $\overline{\kappa_0}$ size decreases from 0.049 s to 0.020 s. This phenomenon is in line with the common understanding, which the hard site has a lower $\kappa_0$ than soft site [26].

## Validation of the region-specific parameters

At Beijing time on August 6, 2023, the Dezhou city in North China witnessed an *M* 5.5 earthquake. The CSMNC and USGS reported the epicenter was 37.16°N, 116.34°E and the focal depth was about 18 km. The maximum earthquake intensity was VII, and the total area of regions with VI and VII covered 224 km², mainly including most of the townships in Plain Country. Moreover, most parts of North China felt the tremor, which was typical characteristic of this earthquake. As a result of the earthquake, 24 people suffered minor injuries and 213 houses were damaged. The focal mechanism solution suggested that this earthquake was strike-slip fault type. In terms of seismic tectonics, the Dezhou earthquake occurred on the western edge of Tan-Lu Fault Zone, which was one of the more active seismic activity zones in the hinterland of North China Plain.

In this study, we make some comparisons between the finite-fault modeling in North China (EXSIM-NC) and previous simulation modeling (EXSIM12). The Dezhou earthquake is taken as an example for the validation of the region-specific key parameters obtained in this paper.

## Model parameters and bias

The model parameters required for this simulation used by both models are summarized in Table 3. The strike and dip of fault plane from far-field body wave inversion by USGS is adopted. The fault plane dimension is estimated by the empirical relationship between the size of fault and the moment magnitude given by Wells and Coppersmith [27]. In this paper, main rupture fault plane is divided into 7 × 5 subfaults with subfault size of 1 km × 1 km. Owing to relatively small magnitude of Dezhou earthquake, the random distribution of slip is used to simulate source rupture process. The stress drop of both models are 60 bars by the trial-and-error method mentioned in the next section. The crustal density in source region and shear-wave velocity are taken as 2.8 g/cm³ and 3.5 km/s from the study of Xu [25]. The other source parameters, rupture velocity and pulsing area percentage are 0.8 times of shear-wave velocity and 50%, according to the recommendations

**Table 3. Stochastic finite-fault simulation input parameters of model EXSIM-NC and model EXSIM12 for Dezhou earthquake.**

| Model parameter | Parameter value | |
|---|---|---|
| Fault strike and dip (deg) | 37,70 | |
| Fault length and width (km) | 7×5 | |
| Slip distribution | Random distribution | |
| Subfault dimension (km×km) | 1×1 | |
| Stress drop (bars) | 60 (This study) | |
| Crustal density (g/cm³) | 2.8 | |
| Shear-wave velocity, $\beta$ (km/s) | 3.5 | |
| Rupture speed (km/s) | 0.8×$\beta$ | |
| Pulsing area percentage | 50% | |
| Geometric spreading $G(R)$ | $1/R$ ($R{\le}40$ km), $1/R^{0.5}$ ($R{>}40$ km) | |
| Distance-dependent duration $T_d$ | 0 ($R{<}10$ km), $T_0+0.05R$ ($R{\ge}10$ km) | |
| Quality factor $Q(f)$ | $248.0f^{0.70}$ | |
| Crustal site amplification | Amplification for generic rock site | |
| Local site amplification | EXSIM-NC: Class III (This study) | EXSIM12: Class C |
| $\kappa_0$ (s) | EXSIM-NC: 0.049 (This study) | EXSIM12: 0.04 |

of Motazedian and Atkinson [17]. The path attenuation effects mainly involve the combination of geometric spreading, ground-motion duration effect and anelastic attenuation. The distance-dependent geometric spreading function adopts the bilinear model, which is expressed in the form of $G(R)=1/R$ for $R \leq 40$ km and $G(R)=R^{0.5}$ for $R>40$ km [28]. The duration model of Atkinson and Boore is adopted [4], which is expressed as $T_d=T_0+0.05 \times R$, where $T_0$ represents the rise time. The anelastic attenuation frequency-dependent $Q(f)$ inferred by Zhao is used in this simulation [29], which is expressed as $Q(f)=248.0f^{0.70}$. In site effect, the crustal amplification factor, which is established based on North America ground motion data by Boore and Joyner [24], is selected. In the stochastic finite-fault method and compute programs, local site amplification and near-surface attenuation are considered separately, but what matters for ground-motion estimation is the combined effect of local site amplification and high-frequency decay. All stations selected in the simulation are located in North China Plain. Therefore, in model EXSIM-NC, the local site effect generated by using the quarter-wavelength method and the high-frequency decay factor $\kappa_0$ calculated by the spectral decay method in preceding sections are adopted, respectively. As a comparison, in model EXSIM12, local site effect of Site Class C in the NEHRP building code proposed by Boore and Joyner is selected and high-frequency decay factor $\kappa_0$ is fixed at 0.04 s found by Anderson and Hough at the same time [21,24]. Thus, the main difference between model EXSIM-NC and model EXSIM12 is the different values of the combined effect of amplification and attenuation.

In addition, the location of 5 fixed stations that are used for validation, namely FXT, QFT, XXT, YJT and LYT, is depicted in Fig 1, and detailed information related to the 5 selected sites is listed in Table 2.

Stress drop parameter $\Delta\omega t\sigma$ is the most important parameter controlling the high-frequency spectral amplitude and high-frequency energy content. The actual stress drop of earthquake is sometimes unknown, this study adopts the trial-and-error approach to estimate $\Delta\omega t\sigma$ in the case of Dezhou earthquake. The model bias for each station can be defined as follows:

$$B(f) = \log(Sim./Obs.) = \log(PSAsim(f)/\sqrt{PSA_{EW}(f) \times PSA_{NS}(f)}) \tag{7}$$

where $PSA_{sim}(f)$ is the average PSA of simulation results for each site, and $\sqrt{PSA_{EW}(f) \times PSA_{NS}(f)}$ is the geometric mean of PSA recorded by EW and NS horizontal components at the same site [11].

All simulated parameters other than $\Delta\omega t\sigma$ required for simulating the Dezhou earthquake are presented in Table 3. The Dezhou earthquake event is recorded by 5 stations in this study, which are included in Table 2, namely FXT, QFT, XXT, YJT and LYT. In this section, the simulated average PSA of 5 fixed stations are computed by using the stress drop value of 30 bars, 40 bars, 50 bars, 60 bars, 70 bars, 80 bars, 90 bars and 100 bars, respectively, and model bias is calculated, as depicted in Fig 4. Overall, when the stress drop is set to 60 bars, the error between simulated and recorded PSA is minimized. In addition, it is worth mentioning that trial-and-error method is also applicable to determination of other model parameters.

By employing the above mentioned parameters, acceleration time histories, peak ground acceleration, Fourier amplitude spectrum and 5%-damped pseudo-acceleration response spectrum of 5 strong motion stations selected in the study are calculated using two models, respectively.

Here, model bias is defined as logarithm of the ratio of simulated to observed PSA. Similarly, observed PSA is expressed as the geometric mean value of two orthogonal horizontal components, namely EW and NS. The plots of mean modeling bias versus frequency for Dezhou earthquake using two approaches are illustrated in Fig 5. It is observed that the model EXSIM-NC has a bias close to 0 in the periods ranging from 0.01 s to 0.4 s and from 4 s to 10 s, which is better as compared to model EXSIM12 for the same periods. Bias of the model EXSIM-NC remains in the range between ±0.26 log units at periods from 0.1 s to 10 s, indicating that PSA could be generally well matched by the model EXSIM-NC in the period range. With regard to previous method, the bias at medium-long period (T>1 s) is relatively larger than model EXSIM-NC, and the maximum bias is about −0.35 log units. In other words, the simulated PSA are about 0.45 times that

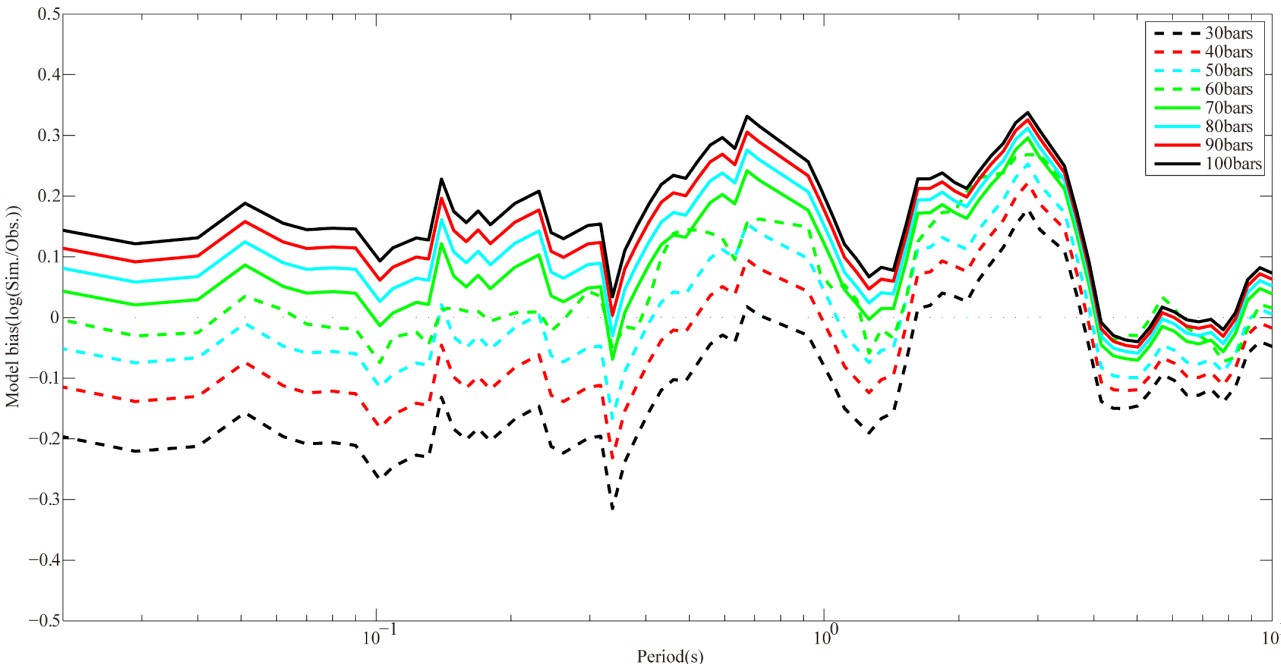

**Fig 4. Estimation of stress drops for the 2023 Dezhou earthquake.** Solid lines in different color and line style denote model bias when the stress drop is 30 bars, 40 bars, 50 bars, 60 bars, 70 bars, 80 bars, 90 bars and 100 bars, respectively.

of the observed result. Overall, the results show that the model EXSIM-NC has higher accuracy as compared to previous method (EXSIM12). This validates the effectiveness of region-specific key parameters we obtained in this study.

## Comparison of PGA and acceleration time series

Observed recordings in EW and NS components and simulated acceleration time series at the selected stations obtained from the two models are displayed in Fig 6, and corresponding PGA values of these recordings are shown in Fig 6. It can be seen from the comparison results that simulated PGA of model EXSIM-NC are in good agreement with actual PGA values at most of the selected stations except station LYT, where the PGA is slightly overestimated by simulation of model EXSIM-NC. However, the PGA values are significantly underestimated by simulation of model EXSIM12 with the exception of station LYT. In addition, as can be observed from Fig 1 and Table 2, station LYT is located in mountains areas and bedrock site. Therefore, discrepancies in PGA may be related to irregular topography and local site type [30]. If more detailed information with regard to topographic and site conditions is completely taken into consideration, the simulation accuracy of station LYT would be further improved.

On the other hand, comparison results also present that the duration of simulated times series are shorter than the real duration of observed recordings at stations QFT, XXT and YJT, which are located in soil site. Thus, near surface soil layer reflection and path propagation effects have a significant impact on ground motion duration. The result indicates that duration model of Atkinson and Boore underestimates the actual ground motion duration of Dezhou earthquake [4]. Further research is required for obtaining a more reasonable distance-dependent duration model of stochastic simulation method in North China.

## Comparison of FAS and PSA

As can be seen from the Fig 7, we calculate the simulated FAS and the recorded FAS of the 5 selected stations, which shows the synthetic FAS of model EXSIM-NC at most of the stations except station LYT, are matched with the recorded

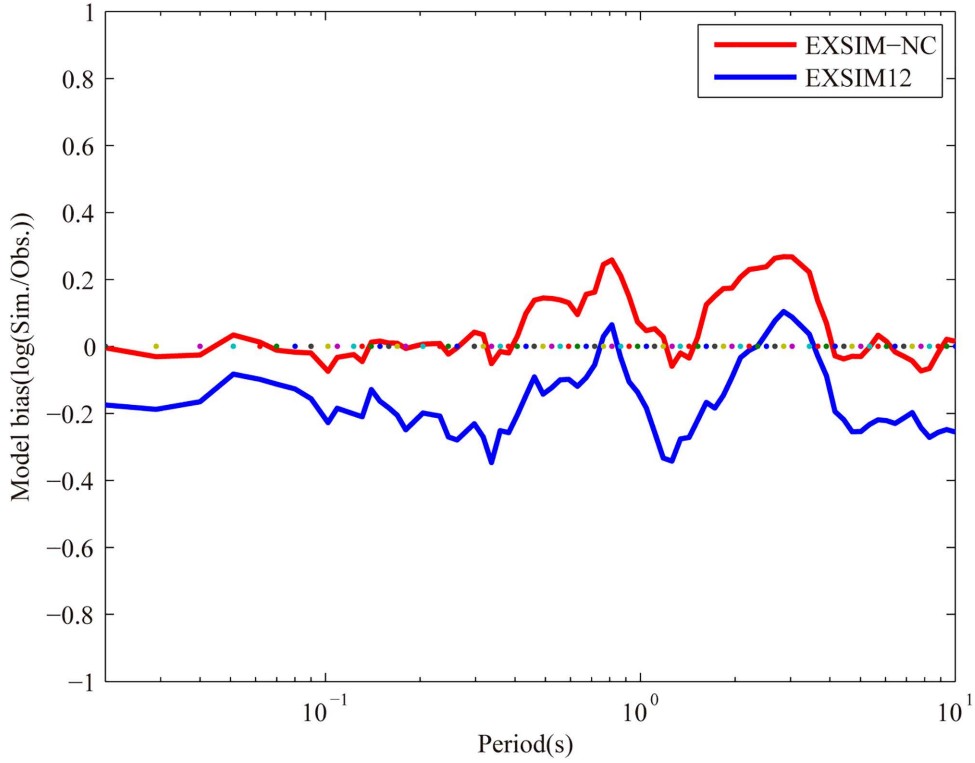

**Fig 5. Bias for modeling results obtained by the finite-fault modeling in North China (EXSIM-NC) and previous method (EXSIM12).** The model bias is calculated at a series of period equally spaced on the logarithmic scale from 0.01 to 10 s.

FAS. At observation stations except LYT, the synthetic FAS have good spectral consistent with the recorded FAS at high frequencies more than 1 Hz, but are obviously underestimated at low frequencies below 1 Hz. The reason for this results may be resulted from the stochastic finite-fault method is mainly utilized for strong ground motion simulation in high frequency band (f > 1 Hz) [17,31].

Further comparisons of the 5%-damped pseudo-spectral acceleration response spectra are made between simulated time series and observed recordings in the EW and NS components at 5 selected stations. In Fig 8, we can observe that the simulated PSA of model EXSIM-NC are well matched with the PSA of recordings except station LYT, where the PSA are obviously overestimated in full period. Moreover, except station LYT, the simulated PSA of model EXSIM12 are quite different from the observed PGA. In fact, the site amplification and decay parameter $\kappa_0$ used in model EXSIM-NC are estimated based on the borehole data and ground motion recordings at North China Plain region, which cannot truly reflect the actual site conditions of the station LYT located. This could possibly explain the inconsistency at station LYT.

Overall, we observe that the FAS and PSA values synthesized by model EXSIM-NC conform sufficiently with the recorded values at most of the selected stations.

## Comparison with PGA contour map

After the comparison of PGA, duration and spectrum, we compare simulated near fault PGA distribution with shake-map of PGA released by USGS for the 2023 Dezhou earthquake (https://earthquake.usgs.gov/earthquakes/eventpage/us6000ky5l/executive).

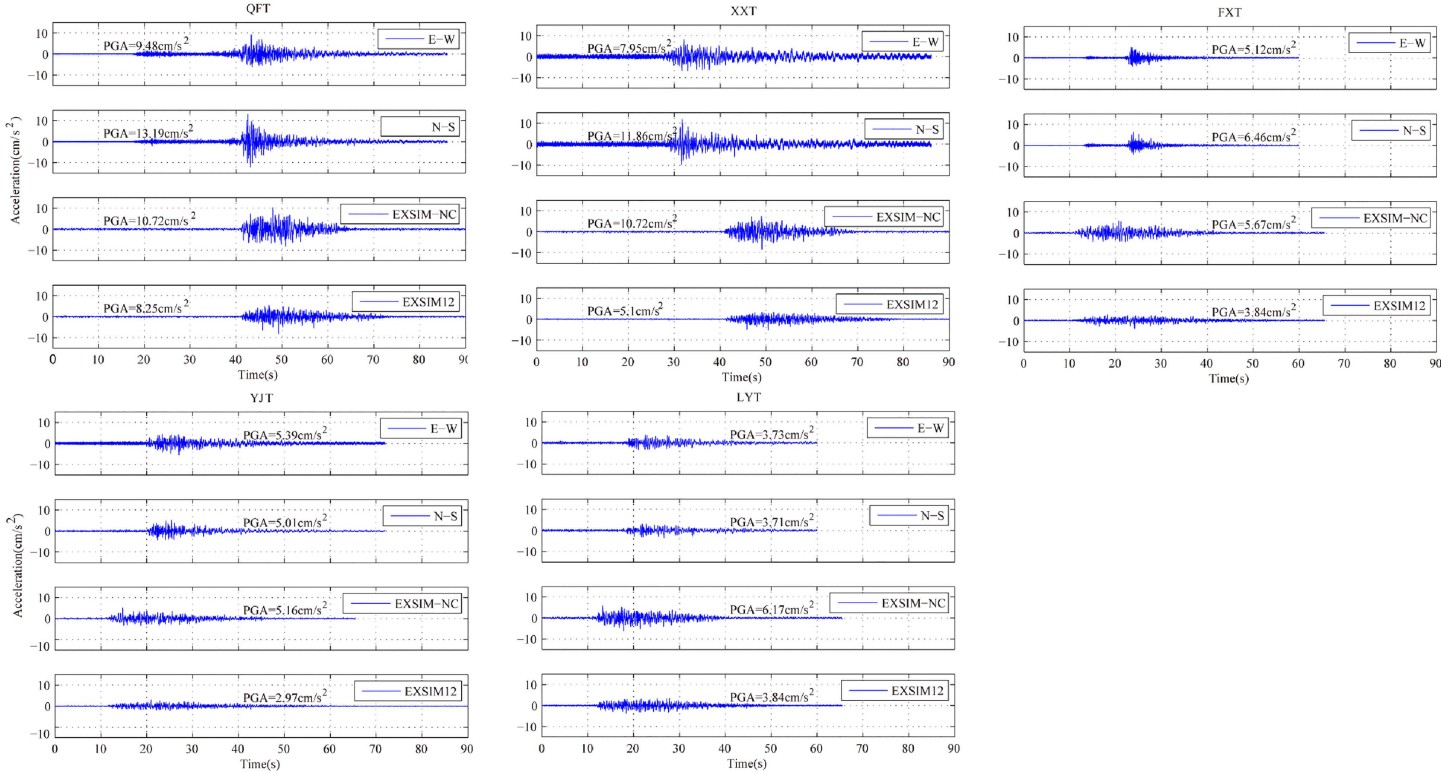

**Fig 6. Comparison between the simulated acceleration time series and actual recordings (EW and NS components) at 5 selected stations.** The peak value of each acceleration time series is presented in the figure.

In order to obtain the simulated PGA contour map, a grid with 42 sites are stablished in the region around the fault, whose ranges of latitude and longitude are given in Fig 9. To improve accuracy, the grid spacing is much denser in the vicinity of the epicenter. In addition, PGA of each site we choose is the average result of 30 runs simulation. Consequently, the simulated PGA contour map of Dezhou earthquake as shown in Fig 9 is inferred from the simulated PGA at each site. From the PGA contour map, we can observe that the maximum PGA reaches close to 250 m/s2 and the largest earthquake intensity is VII degree, which are in a fair agreement with the shakemap of PGA released by USGS. In Fig 9, the dark red area around the epicenter, the light red area and the green area indicate the VII degree, VI degree and V degree earthquake intensity zone, respectively. Furthermore, the yellow circle represents the VI degree isoseismic line, inside where structure and population are more vulnerable to earthquake shaking. Based on the further comparison between Fig 9 and the shakemap released by USGS, we can observe that the size of each intensity zone and the shape of each isoseismic line in this study are in high agreement with the the research of USUS.

From the above comparison, we can conclude that the finite-fault modeling in North China proposed in this study could generally better fit the characteristic of strong ground motion of the Dezhou earthquake, compared with the previous method (EXSIM12). The validation of region-specific key parameters demonstrates that it could be applied to the simulation of high-frequency ground motion in North China.

## Discussion and conclusions

In this paper, we review and modify the stochastic finite-fault method based on dynamic corner frequency (EXSIM12). Based on the borehole data and ground motion recordings, some region-specific seismic parameters including site

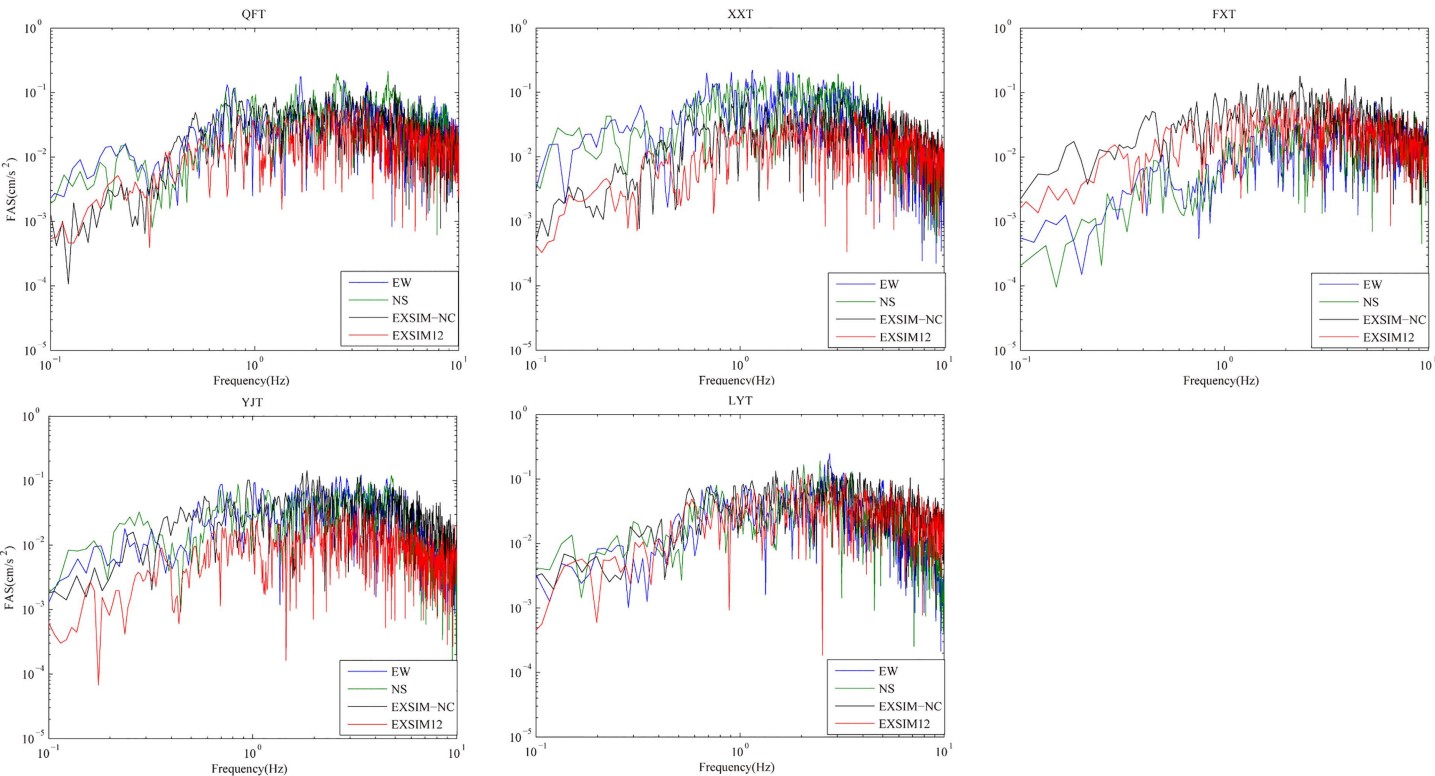

**Fig 7. Comparison of simulated and observed FSA at 5 selected stations.** The observed FSA of EW and NS components are indicated with blue and green solid lines, and the simulated FSA by model EXSIM-NC and model EXSIM12 are indicated with black and red solid lines, respectively.

amplification, kappa and stress drop in North China region are estimated and analyzed in our study. To validate the finite-fault modeling in North China (EXSIM-NC), we calculate the model bias and compare the ground motion results for the Dezhou earthquake obtained by our study with those by the previous method.

The main conclusions are as follows:

The local site amplification for each station and the mean amplification values versus frequency for class II and III site are estimated by using the quarter-wavelength approximation, respectively. The result demonstrates that the shear-wave velocity and sedimentary thickness have a significant effect on the size of amplification factor.

(2)   The high-frequency decay parameter $\kappa_0$ is calculated in North China, the result of which suggests the mean $\kappa_0$ of 0.049 s in Plain region and $\kappa_0$ of 0.02 s in Mountain region, respectively. The result implies that $\kappa_0$ has strong regional characteristics and is severely affected by the local site conditions. In our study, there is an obvious decreasing trend in the size of $\kappa_0$ as the site condition changes from sedimentary plain to mountain area.

(3)   The trial-and-error approach is adopted to estimate stress drop for Dezhou earthquake. For the determination of other uncertain input parameters, we recommend to consider using this method.

(4)   By employing the appropriate input parameters, the finite-fault modeling in North China (EXSIM-NC) and previous simulation modeling (EXSIM12) are used to synthesize acceleration time histories of Dezhou $M$ 5.5 earthquake, respectively. Comparison of model bias demonstrates that PSA could be generally well matched by the model EXSIM-NC. The simulated PGA, FSA and PSA of most stations obtained by model EXSIM-NC are in good agreement

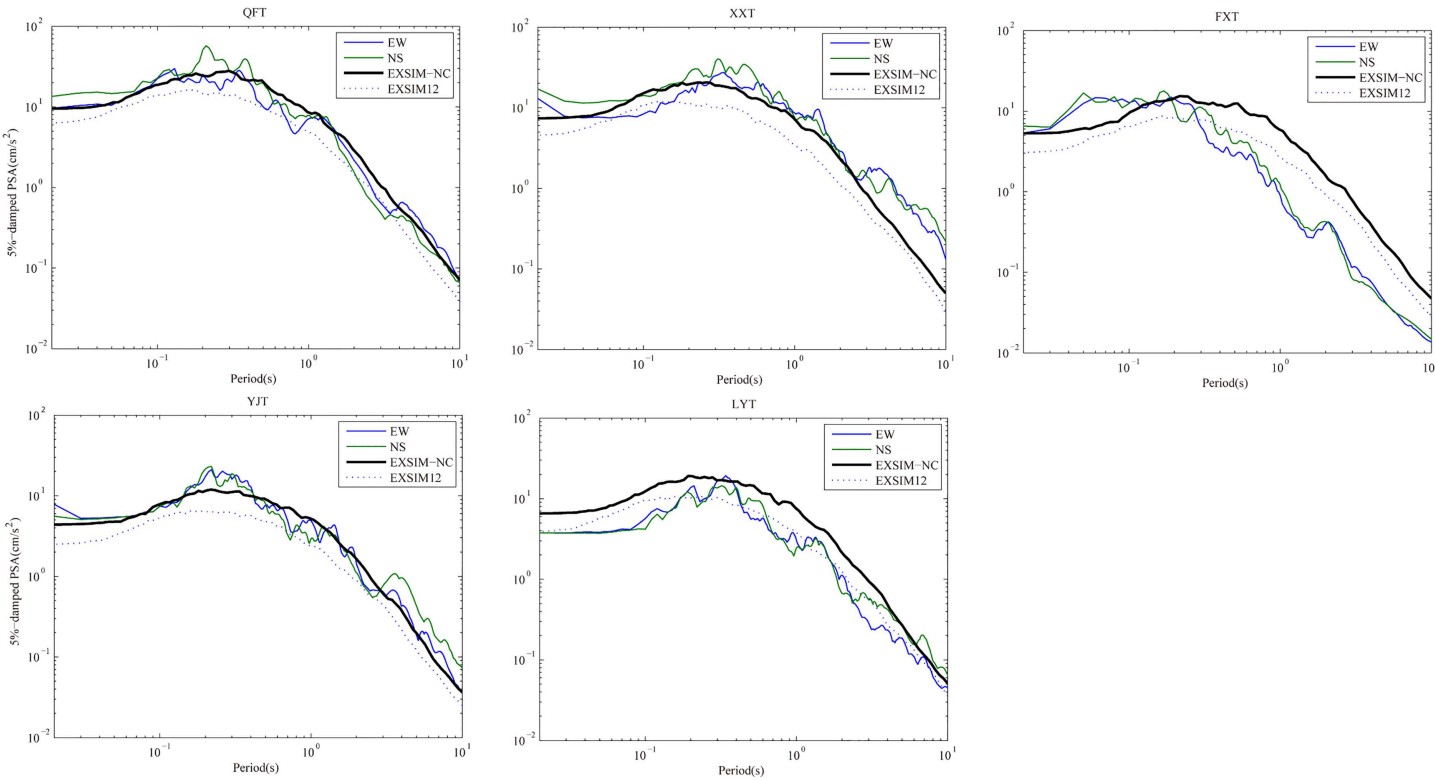

**Fig 8. Comparison of simulated and observed 5%-damped pseudo-spectral acceleration at 5 selected stations.** The observed PSA of EW and NS components are indicated with blue and green solid lines, and the simulated PSA by model EXSIM-NC and model EXSIM12 are indicated with black solid and dashed lines, respectively.

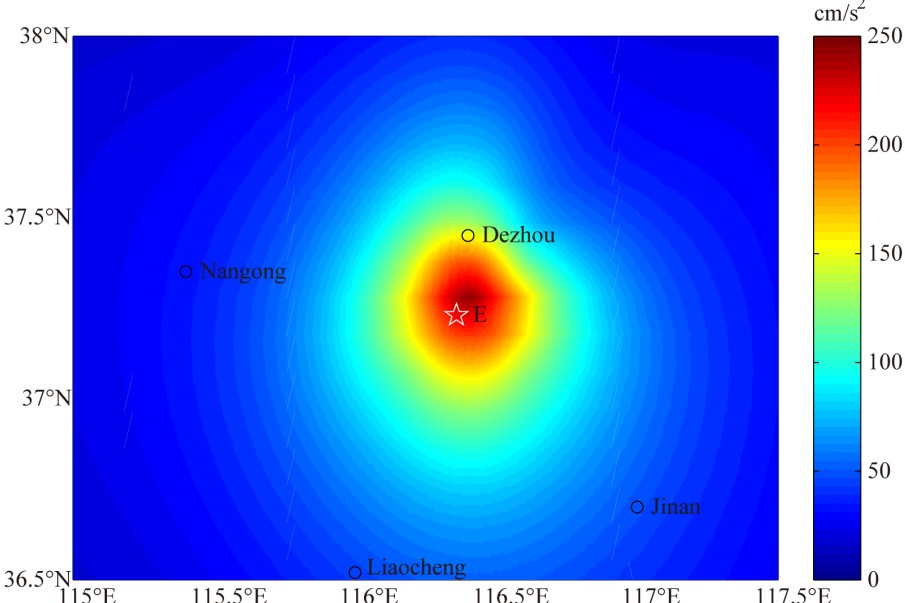

**Fig 9. The simulated PGA field of the 2023 Dezhou earthquake in this study.** The contours indicate PGA in cm/s². The epicenter is marked as the star and the locations of some cities are indicated by circle.

with those observed, except station LYT. The discrepancies may be related to irregular topography and actual local site condition where station LYT is located on. In addition, the simulated PGA field is in consistent with the the shakemap of PGA released by USGS. Finally, the effectiveness of region-specific key seismic parameters is validated. As described by Motazedian and Atkinson [17], the advantage of the stochastic method are the good performance at high frequencies. Furthermore, the region-specific parameters and model EXSIM-NC proposed in this study can be recommended to synthetic high-frequency ground motion in North China.

## Supporting information

**S1 File. Acceleration time history data.** The Strong motion recordings for this study are provided by China Strong Motion Network Centre at Institute of Engineering Mechanics, China Earthquake Administration.
(ZIP)

**S2 File. Program code.** The program code of strong ground motion simulation based on stochastic finite-fault method with dynamic corner frequency for the Pingyuan earthquake.
(ZIP)

## Acknowledgments

The authors are grateful to Atkinson, Boore and Motazedian for sharing the programs of simulating ground motions. We would like to thank USGS for providing earthquake information and China Strong Motion Network Centre for providing observed acceleration time histories. Last but not least, we sincerely thank two anonymous reviewers for their helpful comments and advise that improve the article.

## Author contributions

**Conceptualization:** Xiaoshan Wang.

**Data curation:** Xiaohui Jia, Zihan Feng.

**Formal analysis:** Xiaohui Jia.

**Resources:** Aiwen Liu.

**Software:** Xiaoshan Wang.

**Supervision:** Xiaoshan Wang, Aiwen Liu.

**Validation:** Xiaoshan Wang, Aiwen Liu.

**Visualization:** Aiwen Liu.

**Writing – original draft:** Xiaohui Jia.

**Writing – review & editing:** Xiaohui Jia, Aiwen Liu.

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
