## [Decision Letter · Decision Letter 0]

31 Jul 2025

Dear Dr. Jia,

Thank you for submitting your manuscript to PLOS ONE. After careful consideration, we feel that it has merit but does not fully meet PLOS ONE’s publication criteria as it currently stands. Therefore, we invite you to submit a revised version of the manuscript that addresses the points raised during the review process.

We look forward to receiving your revised manuscript.

Kind regards,

Benedetto Schiavo, Ph.D.

Academic Editor

PLOS ONE

Journal Requirements:

[Doctoral Research Initiation Foundation of Hebei GEO University under Grant Nos. 2024074]. 

[This study is supported by the National Natural Science Foundation of China under Grant Nos. 52278540 and Doctoral Research Initiation Foundation of Hebei GEO University under Grant Nos. 2024074. The authors are grateful to Atkinson, Boore and Motazedian for sharing the programs of simulating ground motions. We would like to thank USGS for providing earthquake information and China Strong Motion Network Centre for providing observed acceleration time histories. Last but not least, we sincerely thank three anonymous reviewers for their helpful comments and advise that improve the article.]

[Doctoral Research Initiation Foundation of Hebei GEO University under Grant Nos. 2024074]

4. We note that Figure 1 in your submission contains map images which may be copyrighted. All PLOS content is published under the Creative Commons Attribution License (CC BY 4.0), which means that the manuscript, images, and Supporting Information files will be freely available online, and any third party is permitted to access, download, copy, distribute, and use these materials in any way, even commercially, with proper attribution. For these reasons, we cannot publish previously copyrighted maps or satellite images created using proprietary data, such as Google software (Google Maps, Street View, and Earth). For more information, see our copyright guidelines: http://journals.plos.org/plosone/s/licenses-and-copyright.

We require you to either (1) present written permission from the copyright holder to publish these figures specifically under the CC BY 4.0 license, or (2) remove the figures from your submission

5. We are unable to open your Supporting Information file [Support information.zip]. Please kindly revise as necessary and re-upload.

Reviewers' comments:

Reviewer's Responses to Questions

**Comments to the Author**

1. Is the manuscript technically sound, and do the data support the conclusions?

Reviewer #1: Yes

Reviewer #2: Yes

2. Has the statistical analysis been performed appropriately and rigorously?

Reviewer #1: Yes

Reviewer #2: No

3. Have the authors made all data underlying the findings in their manuscript fully available?

Reviewer #1: Yes

Reviewer #2: Yes

4. Is the manuscript presented in an intelligible fashion and written in standard English?

Reviewer #1: No

Reviewer #2: Yes

Reviewer #1: 1. The calculation methods for site amplification, kappa, and stress drop are commonly used in related literature on stochastic finite-fault simulations. It is needed to clarify the novelty of this study.

2. In the introduction of this paper, there are too many descriptions of generality, but there is insufficient analysis of the current research.

3. The station YJS appears twice in Table 2, and the coordinates of station AYS in Table 2 differ from its location shown in Figure 1. It is needed to verify the station information in Table 2.

4. Lines 314–315: Station information is presented in Table 2, not Table 1. Supplementary Material 1 contains strong motion recordings at six stations, but none of them match the five stations (FXS, QFS, XXS, YJS and AYS5) selected in the simulation. Why?

5. The XXS station records in Figure 6 contain significant noise, and necessary processing should be performed on the seismic records.

Reviewer #2: The authors estimated some input parameters, such as site amplifications, high-frequency attenuation and stress drop, in the study area, and simulated the PGA, FAS, acceleration time history, PSA, and shake map of PGA for the Dezhou earthquake. The simulated results coincided well with observed values. However, some issues should be addressed before the manuscript can be published.

1.Lines 39 and 71, peak ground acceleration (PGA)?

2.Line 97, GB50011-201015 ?

3.Equation (2) is not correct. (1+(f/foij)2).

4.Parameter Hij in equation (2) is not defined.

5.Line 155. The authors claimed that some modifications or variation of EXSIM12 on some input model parameters. These are only estimation of regional parameters based on seismic records in the study region, and are not improvements to the EXSIM model. The author should revise the relevant descriptions throughout the entire text.

6.Line 181, region21, ?

7.Equation (6) is not correct, kappa is defined as the slope, which is calculated by the least square method, instead of directly calculating the slope using κ = -(lnA(f2)-lnA(f1))/( π(f2-f1)).

8.Line 229, κ = k/(πloge) may be wrong. If the FAS curve was plotted in a semi-logarithmic space, the parameter κ can be obtained by the following equation κ= k/π. Equation (6) contradicts the equation in line 229.

9.Lines 228 and 230, Kappa should be kappa.

10.Line 295, Motazedian14, ?

11.Line 298, Atkinson and Boore1, ?

12.Figure 3(c) and (d), what is the reason for a negative slope?

13. Line 53, Boore13, ?

**Do you want your identity to be public for this peer review?** For information about this choice, including consent withdrawal, please see our Privacy Policy

Reviewer #1: No

Reviewer #2: No

---

## [Author Response · Author response to Decision Letter 1]

3 Oct 2025

PLOS ONE

Ref: Submission ID PONE-D-25-27507

Title: Parameters estimation of stochastic finite fault ground motion simulation method and its application in North China

Response: Letter to Reviewer’ Comments

Comments from the Editor:

We thank all the editors and reviewers for their valuable comments and suggestions. We have carefully revised the manuscript to enhance its clarify and facilitate the understanding of the readers. Our point-to-point responses are presented in the following. The changes in the revised manuscript have been highlighted. We hope that the revision would satisfactory address the comments and concerns of the editors and reviewers.

1. Thank you for stating the following financial disclosure: [Doctoral Research Initiation Foundation of Hebei GEO University under Grant Nos. 2024074]

Response: What role the funders took in the study have been stated in the manscript.

Funding: This work is financially supported by the Doctoral Research Initiation Foundation of Hebei GEO University (Grant Nos. 2024074). The funders conducted the research, including study design, data collection and analysis, and preparation of the manuscript.

2.Thank you for stating the following in the Acknowledgments Section of your manuscript:

[This study is supported by the National Natural Science Foundation of China under Grant Nos. 52278540 and Doctoral Research Initiation Foundation of Hebei GEO University under Grant Nos. 2024074. The authors are grateful to Atkinson, Boore and Motazedian for sharing the programs of simulating ground motions. We would like to thank USGS for providing earthquake information and China Strong Motion Network Centre for providing observed acceleration time histories. Last but not least, we sincerely thank three anonymous reviewers for their helpful comments and advise that improve the article.]

Response: We have removed any funding-related text from the manuscript.

Funding: “This study is supported by the National Natural Science Foundation of China under Grant Nos. 52278540 and Doctoral Research Initiation Foundation of Hebei GEO University under Grant Nos. 2024074.” have been deleted.

3.We note that Figures 1 in your submission contain [map/satellite] images which may be copyrighted...

Response: In the paper, Figure 1 created by us are plotted by GMT (Generic Mapping Tools), which is an open source 2D geographic data visualization tool and widely used in the earth sciences. Hence, Figure 1 are not previously copyrighted maps or satellite image.

The software used to create the Figure 1 is cited in the figure legend (including version number and URL link).

Figure 1. Distribution of the study earthquakes and strong motion stations in North China study area. The epicenters and selected stations are represented as pentagram and triangle, respectively. Figure 1 is plotted by the Generic Mapping Tools (GMT) version 5.4.4 (https://gmt-china.org/download/).

Reviewer # 1:

1.The calculation methods for site amplification, kappa, and stress drop are commonly used in related literature on stochastic finite-fault simulations. It is needed to clarify the novelty of this study.

Response: Thanks for your positive comments. We have carefully revised this manuscript based on the reviewer’s comments. We hope that the revisions and improvements would satisfactory address the reviewers’ concerns.

The paragraph from line 151 to 160 in Chapt 3 introduced the work we study, which clarify the novelty partly. I hope this could meet your demand.

The above mentioned dynamic corner frequency modeling was proposed by Motazedian and Atkinson [17], and made further improvements on spectral amplitudes by Boore [16]. This study adopted the stochastic finite-fault open computer program in FORTRAN named EXSIM12, which was complied by Atkinson and Assatourians [22]. However, this method needs to build and input source parameters including stress drop and slip distribution, path parameters including quality factor and geometric spreading behavior, and site parameters such as crustal amplification function, local site amplification and . In addition, most of these parameters are often different in specific region. Consequently, the more accurate region-specific input parameters we obtain, the more realistic acceleration time series with the period range of engineering interest we can reproduce.

2.In the introduction of this paper, there are too many descriptions of generality, but there is insufficient analysis of the current research.

Response: The statement in the abstract have been revised.

The comparison results confirm the validity of the updated asperity generation method, which can reflect the effect of actual slip distribution on the variation of source spectrum and avoid inaccurate estimation of the largest PGA.

3.The station YJS appears twice in Table 2, and the coordinates of station AYS in Table 2 differ from its location shown in Figure 1. It is needed to verify the station information in Table 2.

Response: Thanks for your correct comments. I think that such mistakes should be avoided.

The station information in Table 2 and Fig.1 have been verified as follows.

Table 2. Information on the strong motion stations used in this study.

Station Code Lat (N) Lon (E) Site class (China) Site class (NEHRP) Sensor Type

LGZ 39.44 115.36 II B ETNA/ES-T

HMY 41.10 116.89 II B ETNA/ES-T

GLZ 40.71 114.55 II B ETNA/ES-T

JEG 40.14 114.29 II B ETNA/ES-T

ZNB 41.00 115.57 II B ETNA/ES-T

GXC 41.43 115.79 II B ETNA/ES-T

YAJ 39.93 116.83 III C ETNA/ES-T

WBH 39.67 117.05 III C ETNA/ES-T

GGZ 39.61 116.58 III C ETNA/ES-T

BZZ 39.62 116.57 III C ETNA/ES-T

BGZ 39.35 116.66 III C ETNA/ES-T

DCF 39.49 115.79 III C ETNA/ES-T

XHY 40.10 114.80 Soil Soil ETNA/ES-T

ZJP 39.90 115.30 Soil Soil ETNA/ES-T

FHL 40.10 116.10 Soil Soil MR2002/SLJ-100

SSL 40.20 116.30 Soil Soil MR2002/SLJ-100

SDT 40.00 116.20 Soil Soil MR2002/SLJ-100

SDZ 39.60 115.60 Soil Soil MR2002/SLJ-100

STJ 40.00 115.50 Soil Soil ETNA/ES-T

SWD 40.00 116.00 Soil Soil MR2002/SLJ-100

WJY 40.20 116.00 Soil Soil MR2002/SLJ-100

XYT 40.40 115.80 Soil Soil MR2002/SLJ-100

YJS 39.60 115.80 Soil Soil ETNA/ES-T

BTS 40.40 115.20 Soil Soil ETNA/ES-T

DBX 40.10 115.10 Soil Soil ETNA/ES-T

HHY 40.20 115.30 Soil Soil ETNA/ES-T

LAS 40.40 115.70 Soil Soil ETNA/ES-T

DOL 40.20 117.70 Soil Soil ETNA/ES-T

HOQ 39.80 117.70 Soil Soil ETNA/ES-T

WLG 39.70 117.80 Soil Soil ETNA/ES-T

JZG 39.40 118.10 Soil Soil ETNA/ES-T

QFT 35.80 115.00 Soil Soil ETNA/ES-T

XXT 35.00 113.90 Soil Soil ETNA/ES-T

FXT 35.80 115.50 Soil Soil GSMA-24IP/SLJ-100

YJT 35.10 114.20 Soil Soil ETNA/ES-T

LYT 34.30 112.20 Rock Rock GSMA-24IP/SLJ-100

4.Lines 314–315: Station information is presented in Table 2, not Table 1. Supplementary Material 1 contains strong motion recordings at six stations, but none of them match the five stations (FXS, QFS, XXS, YJS and AYS5) selected in the simulation. Why?

Response: Thank you for pointing this out.

In the lines 314-315, the Table 1 have verified into Table 2.

In the revised paper, the five stations (FXT, QFT, XXT, YJT and LYT) have been renamed, which match with the Supplementary Material 1.

5. The XXS station records in Figure 6 contain significant noise, and necessary processing should be performed on the seismic records.

Response: Thanks for the valuable comment.

The XXT station records have been checked and processed again, and we replot the new picture with little noise.

It should be noted that strong motion recordings for this study are provided by China Strong Motion Network Centre at Institute of Engineering Mechanics, China Earthquake Administration.

The CSMNC declares that all recordings have been filtered and baseline-corrected. I will contact the CSMNC to discuss the noise. In addition, we will check the records in our study ourselves.

In fact, the significant noise has little influence on PGA analysis and spectrum analysis in next section.

Reviewer # 2:

The authors estimated some input parameters, such as site amplifications, high-frequency attenuation and stress drop, in the study area, and simulated the PGA, FAS, acceleration time history, PSA, and shake map of PGA for the Dezhou earthquake. The simulated results coincided well with observed values. However, some issues should be addressed before the manuscript can be published.

Response: Thanks for the valuable comment. We appreciate your summary of the manuscript and encouraging comment. Thanks for your positive comments. We have carefully revised this manuscript based on the reviewer’s comments. We hope that the revisions and improvements would satisfactory address the reviewers’ concerns.

1.Lines 39 and 71, peak ground acceleration (PGA)?.

Response: Thanks for your valuable comments.

The word “PGA” have been deleted in lines 39 and 71.

2.Line 97, GB50011-201015 ?

Response: We think this is an excellent suggestion. The sentence has been modified as follows:

the China Seismic Code GB50011-2010 [15]

3.Equation (2) is not correct. (1+(f/foij)2).

Response: Thanks for the valuable comment. The sentence has been modified as follows:

4.Parameter Hij in equation (2) is not defined.

Response: Thanks for your suggestion. The Hij is the scaling factor to the conserve the high-frequency spectral level.

Below the equation (2), we have introduced the Hij.

In the equation (1) and (2), , , and are the ijth subfault seismic moment, distance from the observation point, scaling factor to conserve the high-frequency spectral level and dynamic corner frequency, respectively.

5.Line 155. The authors claimed that some modifications or variation of EXSIM12 on some input model parameters. These are only estimation of regional parameters based on seismic records in the study region, and are not improvements to the EXSIM model. The author should revise the relevant descriptions throughout the entire text.

Response: Thanks for the valuable comment and accurate comments. As the reviewer mentioned, we did not modify the EXSIM12 program in essence. The most work of our study is to determine the input parameters for stochastic finite-fault ground-motion simulation in North China.

Thus, we modified the inaccurate statement “ improved method” into “improved model” in the revised manuscript.

6.Line 181, region21, ?

Response: We think this is an excellent suggestion. The phrase and sentence has been modified as follows:

Region [21],

7.Equation (6) is not correct, kappa is defined as the slope, which is calculated by the least square method, instead of directly calculating the slope using κ = -(lnA(f2)-lnA(f1))/( π(f2-f1)).

Response: Thanks for your insightful and valuable suggestion. The Equation (6) have been modified as follows.

In our paper, the kappa is exactly calculated by the least square method.

(6)

8.Line 229, κ = k/(πloge) may be wrong. If the FAS curve was plotted in a semi-logarithmic space, the parameter κ can be obtained by the following equation κ= k/π. Equation (6) contradicts the equation in line 229.

Response: Thanks for your valuable comments. The case study, Xingtai earthquake occurred in 1966, and no strong ground stations were built until 2010.

The reason why the Xingtai earthquake was chosen as a case study, although the 1966 Xingtai earthquake lacked strong motion recordings, is because this study was funded by Doctoral Research Initiation Foundation of Hebei GEO University. The funding focuses on earthquakes in the Hebei region. The largest earthquake occurred in recent years is the 2021 Guye Ms5.1 earthquake in Hebei, which was not large enough in magnitude to generate a asperity.

As the saying goes, the case is also the generals chosen from amongst the dwarves.

9.Lines 228 and 230, Kappa should be kappa.

Response: Thank you for pointing this out.

The Kappa have been verified into kappa in the full paper.

10.Line 295, Motazedian14, ?

Response: Thanks for your valuable comments. The sentence has been modified as follows.

according to the recommendations of Motazedian [14]

11.Line 298, Atkinson and Boore1 ?

Response: Thank you for pointing this out. The phrase and sentence has been modified as follows.

The duration model of Atkinson and Boore is adopted [1]

12.Figure 3(c) and (d), what is the reason for a negative slope?

Response: Thanks for the valuable comment and insightful suggestion.

The distribution of kappa versus distance is linear fitted again. A positive slope are obtained in the revised manuscript as shown in Figure 3.

13.Line 53, Boore13, ?

Response: Thank you for pointing this out. The phrase and sentence has been modified as follows.

Stochastic method of simulation (SMSIM) was first proposed by Boore [13]

---

## [Decision Letter · Decision Letter 1]

20 Oct 2025

Dear Dr. Jia,

Thank you for submitting your manuscript to PLOS ONE. After careful consideration, we feel that it has merit but does not fully meet PLOS ONE’s publication criteria as it currently stands. Therefore, we invite you to submit a revised version of the manuscript that addresses the points raised during the review process.

Both reviewers acknowledge the effort made in revising the manuscript, but they still identify important issues that need to be addressed before the paper can be reconsidered for publication.

The reviewers emphasize that the study does not introduce fundamental improvements to the stochastic finite-fault method, but rather focuses on the regional calibration of parameters such as site amplification, kappa, and stress drop. Therefore, the title and text should accurately reflect the nature of this contribution, avoiding the use of expressions such as “improved method” or “improved model.”

Additional clarification is required regarding the observed linear increase of the amplification coefficient with frequency in logarithmic coordinates, as well as the methodology used for the estimation of kappa. The equations presented in the manuscript appear inconsistent, and it should be clearly stated which formulation was used, ensuring that the correct expression is applied throughout. The differences observed in the fitted lines of Figure 3 also need to be explained.

Reviewer 2 notes that some comments from the first review round (specifically comments 8, 9, and 12) remain insufficiently addressed. Please review these points carefully and provide a detailed response. It is also recommended that you discuss your results in the context of recent studies that have advanced the EXSIM methodology, explaining how your regional calibration complements rather than replaces those developments.

We look forward to receiving your revised manuscript.

Kind regards,

Benedetto Schiavo, Ph.D.

Academic Editor

PLOS ONE

Journal Requirements:

Reviewers' comments:

Reviewer's Responses to Questions

**Comments to the Author**

Reviewer #1: All comments have been addressed

Reviewer #2: (No Response)

2. Is the manuscript technically sound, and do the data support the conclusions?

Reviewer #1: Partly

Reviewer #2: Partly

3. Has the statistical analysis been performed appropriately and rigorously?

Reviewer #1: Yes

Reviewer #2: No

4. Have the authors made all data underlying the findings in their manuscript fully available?

Reviewer #1: Yes

Reviewer #2: Yes

5. Is the manuscript presented in an intelligible fashion and written in standard English?

Reviewer #1: Yes

Reviewer #2: Yes

Reviewer #1: 1. Line 272: This study does not introduce fundamental improvements to the stochastic finite-fault method. Instead, it calibrates regional parameters such as local site amplification, high-frequency attenuation factor kappa, and stress drop based on borehole data and records from strong-motion stations. The calibrated parameters are then used for simulations, thereby improving the simulation accuracy. Therefore, we strongly suggest that the authors reconsider whether the current title accurately reflects the work.

2. Lines 187–189: Why does the site amplification coefficient obtained in this study exhibit a linear increase with frequency in logarithmic coordinates? Please clarify the underlying mechanism responsible for this phenomenon.

Reviewer #2: The authors have addressed most of the comments. However, the manuscript still has serious issues that need to be clarified by the authors.

1.The comments 8, 9 and 12 suggested in the first round is not addressed completely.

2.In the Figure 3, the kappa values calculated from records are the same, but the linear fitted lines are different, why?

3.The authors claim that they replaced “improved method” with “improved model”, but I believe that the improved model is also inappropriate. Why does the word “improved” have to be used when the author has not made any improvements to the method or parameter model? Some improvements to the EXSIM method have been made in the following papers, these may be helpful to you.

(1) Slip-correlated high-frequency scaling factor for stochastic finite-fault modeling of ground motion. Bulletin of the Seismological Society of America, 2022, 112(3):1472-1482.

(2) Simulation of earthquake ground motion via stochastic finite-fault modeling considering the effect of rupture velocity. Stochastic Environmental Research and Risk Assessment, 2023, 37:2225-2241.

(3) An updated stochastic finite fault modeling: Application to the Mw 6.0 earthquake in Jiashi, China. Soil Dynamics and Earthquake Engineering, 2022, 162:107450.

The most serious problem is the kappa estimation. In the manuscript, equation (6) and equation defined in line 238 were mentioned. Which one was used? In addition, equation defined in line 238 is completely wrong, which should be κ = –k/(πloge) and was used in cartesian coordinate system with data [A, f]. Equation (6) was used in the semi-logarithmic coordinate system with data [lnA, f].

**Do you want your identity to be public for this peer review?** For information about this choice, including consent withdrawal, please see our Privacy Policy

Reviewer #1: No

Reviewer #2: No

---

## [Author Response · Author response to Decision Letter 2]

3 Nov 2025

PLOS ONE

Ref: Submission ID PONE-D-25-27507

Title: Parameters estimation of stochastic finite fault ground motion simulation method and its application in North China

Response: Letter to Reviewer’ Comments

Comments from the Editor:

We thank all the editors and reviewers for their valuable comments and suggestions. We have carefully revised the manuscript to enhance its clarify and facilitate the understanding of the readers. Our point-to-point responses are presented in the following. The changes in the revised manuscript have been highlighted. We hope that the revision would satisfactory address the comments and concerns of the editors and reviewers.

1. Both reviewers acknowledge the effort made in revising the manuscript, but they still identify important issues that need to be addressed before the paper can be reconsidered for publication.

Response: Thanks for your positive comments.

As can be seen in the Revised Manuscript with Track Changes, we have carefully revised this manuscript based on the important issues and other issues reviewers identified. We hope that the revisions and improvements would address the reviewers’ concerns satisfactorily.

2.The reviewers emphasize that the study does not introduce fundamental improvements to the stochastic finite-fault method, but rather focuses on the regional calibration of parameters such as site amplification, kappa, and stress drop. Therefore, the title and text should accurately reflect the nature of this contribution, avoiding the use of expressions such as “improved method” or “improved model.”

Response: We fully agree with the reviewers’ assessment. As the reviewers mentioned, this paper focuses on some region-specific key parameters including local site amplification, high-frequency decay factor kappa and stress drop. In fact, we have done little work on the essential and fundamental improvements to the stochastic finite-fault method.

According to the reviewers’ advice, the expressions such as “improved method” or “improved model” have been deleted or changed in full paper. On the basis of reading and revising the entire manuscript, the expressions such as “improved method” or “improved model” have been revised.

Please refer to the revised manuscript for details on multiple revisions.

3.Additional clarification is required regarding the observed linear increase of the amplification coefficient with frequency in logarithmic coordinates, as well as the methodology used for the estimation of kappa. The equations presented in the manuscript appear inconsistent, and it should be clearly stated which formulation was used, ensuring that the correct expression is applied throughout. The differences observed in the fitted lines of Figure 3 also need to be explained.

Response: Thanks for the valuable comment.

The observed linear increase of the amplification coefficient with frequency in logarithmic coordinates has clarified in the following text. In logarithmic coordinates, the line is plot form the ten points. The phenomenon of linear increase with frequency are attributed to the limited number of point observations. The linear increase line segment is the line connecting from two adjacent points.

In the methodology used for the estimation of kappa, we used the equation (6), rather than equation defined in line 238. We have deleted the sentence in line 238.

The differences observed in the fitted lines of Figure 3 has been explained and carefully clarified in the in section “Reviewer # 2”.

By checking our drawing data and drawing programs in Matlab, we find that the linear fitted lines and negative slope in original Figure 3(c) and (d) are false. The figure 3 we presented is the original research picture.

We are so sorry that I forget to provide new picture in the manuscript.

The data of κ value and program are checked and verified. Based on more κ value from horizontal components of all recordings, we re-fitted the data points linearly and obtained a linear form in research, but we did not provide these.

The new figure 3 we plot are given as the following, and we provide these in the second round revision.

In the new figure 3 ,the kappa value are re-calculated and the line are re-plot. The new linear fitted lines are given.

4.Reviewer 2 notes that some comments from the first review round (specifically comments 8, 9, and 12) remain insufficiently addressed. Please review these points carefully and provide a detailed response. It is also recommended that you discuss your results in the context of recent studies that have advanced the EXSIM methodology, explaining how your regional calibration complements rather than replaces those developments.

Response: We were really so sorry for our careless mistakes in the first review round. As the reviewer’ mentioned, the comments 8, 9 and 12 were addressed incompletely and insufficiently. For these comments, detailed responses and modifications have been made in the following text, as can be seen in section “Reviewer # 2”.

Compare to the literature that have advanced the EXSIM methodology, our work mainly concentrated on the region-specific key parameters in North China. In deed, we acknowledged that we have did little work on the development or improvement of EXSIM methodology. In fact, this important issue perfectly coincides with the assessment 2 from reviewer 1. Regarding this issue, we have explained our regional parameters calibration rather than method developments based on reading the entire text thoroughly. Please refer to the revised manuscript for details on multiple revisions.

5.To ensure your figures meet our technical requirements, please review our figure guidelines: https://journals.plos.org/plosone/s/figures 

Response: In this review round, in order to make the figures can meet the journal technical requirements, we have read the figure guidelines from the website carefully.

And, to ensure our figures in the paper meet publication quality, figure tool NAAS was adopted to process the original figures.

Finally, the original images and compliant figures in the paper are provided simultaneously.

In addition, the original images were created in a PDF file named ‘S1_raw_images’ as supporting information files.

Reviewer # 1:

1.Line 272: This study does not introduce fundamental improvements to the stochastic finite-fault method. Instead, it calibrates regional parameters such as local site amplification, high-frequency attenuation factor kappa, and stress drop based on borehole data and records from strong-motion stations. The calibrated parameters are then used for simulations, thereby improving the simulation accuracy. Therefore, we strongly suggest that the authors reconsider whether the current title accurately reflects the work.

Response: Thanks for your positive comments. We have carefully revised this manuscript based on the reviewer’s comments.

We fully agree with the reviewers’ assessment. As the reviewers mentioned, this paper does not essentially introduce fundamental improvements in method itself. In fact, we focus on some region-specific key parameters including local site amplification, high-frequency decay factor kappa and stress drop in North China. In addition, the parameters are used to simulate and improve the simulation accuracy. We have reconsidered the title and some expressions to actual reflect our work.

According to the reviewers’ advice, the expressions such as “improved method” or “improved model” have been deleted or changed in full paper. On the basis of reading and revising the entire manuscript, the expressions such as “improved method” or “improved model” have been revised.

Please refer to the revised manuscript for details on multiple revisions.

The title and sentence in Line 272 have been modified as follows:

Validation of the region-specific parameters

In this study, we make some comparisons between the finite-fault modeling in North China (EXSIM_NC) and previous simulation method (EXSIM_12). The Dezhou earthquake is taken as an example for the validation of the region-specific key parameters obtained in this paper.

2.Lines 187–189: Why does the site amplification coefficient obtained in this study exhibit a linear increase with frequency in logarithmic coordinates? Please clarify the underlying mechanism responsible for this phenomenon

Response: Thanks for the valuable comment.

In our study, the local amplification function over versus frequency for class II and III site are computed using the quarter-wavelength approximation.

For a particular frequency, the amplification value is calculated by the square root of ratio between the seismic impedance of bedrock at the depth of source and the average seismic impedance from surface to a quarter wavelength, where seismic impedance is defined as shear wave velocity times density. The algorithm expression is the following:

(4)

where , represent density and shear wave velocity at source, and , represent average density and shear wave velocity at site, respectively.

The quarter-wavelength approximation early used in the following paper, and we refer to this literature in our study.

[24] Boore DM, Joyner WB. Site amplifications for generic rock sites. Bulletin of the Seismological Society of America. 1997; 87(2): 327–341.

As computed and described in the literature [24], the amplification value versus frequency in the literature [24] were presented in the following picture.

In our study, based on the borehole shear wave velocity profile versus depth in North China, we calculated the the amplification value versus frequency. Our research results in North China are as following and can be seen in the Supporting Information.

In figure 3, we plot amplification coefficient obtained in this study with frequency by using the data in the red rectangular box. For example, this is the mean amplification function values estimated in North China.

In logarithmic coordinates, the line is plot from the ten points. The phenomenon of linear increase with frequency are attributed to the limited number of point observations. The linear increase line segment is the line connecting between two adjacent points.

We hope that the clarification for this phenomenon would address the reviewers’ concerns satisfactorily.

Reviewer # 2:

The authors have addressed most of the comments. However, the manuscript still has serious issues that need to be clarified by the authors.

Response: Thanks for the positive comment. We appreciate your summary of the manuscript and encouraging comment. Thanks for your positive comments. We have carefully revised this manuscript based on the reviewer’s comments. We hope that the revisions and improvements would satisfactory address the reviewers’ concerns.

1.The comments 8, 9 and 12 suggested in the first round is not addressed completely.

Response: We were really so sorry for our careless mistakes in the first review round. As the reviewer’ mentioned, the comments 8, 9 and 12 were addressed incompletely and insufficiently. We checked comments in first round and revised carefully as following.

First round 8: 8.Line 229, κ = k/(πloge) may be wrong. If the FAS curve was plotted in a semi-logarithmic space, the parameter κ can be obtained by the following equation κ= k/π. Equation (6) contradicts the equation in line 229.

Response: Thanks for your valuable comments.

In the kappa estimation, we used the equation (6), rather than equation defined in line 229 in first round.

The equation defined in line 229 ( in the first round ) comes from the literature:

(3) An updated stochastic finite fault modeling: Application to the Mw 6.0 earthquake in Jiashi, China. Soil Dynamics and Earthquake Engineering, 2022, 162:107450..

In our paper, the expression of the sentence is referred.

In revised manuscript, the sentence has been deleted as the following:

In Fourier amplitude spectrum diagram, the of one ground motion recording is computed by =k/(loge), where k is the slope and e represents natural constant.

As mentioned by reviewer, we use Equation (6) in the semi-logarithmic coordinate system .

First round 9: 9.Lines 228 and 230, Kappa should be kappa.

Response: Thanks for your valuable comments.

The Kappa have been verified into kappa (κ) in the full paper.

Especially in the section 4.2 kappa (κ), we modified the false Kappa, and verified into κ .

As can be seen in the Revised Manuscript, the section has been modified as follows:

First round 12: 12.Figure 3(c) and (d), what is the reason for a negative slope?

Response: Thanks for your valuable and insightful comments.

By checking our drawing data and drawing programs in Matlab, we find that the linear fitted lines and negative slope in original Figure 3(c) and (d) are false .

We are so sorry that I forget to provide new picture in the manuscript. I think that the negative slope in original picture is false.

New data of κ value and program are collected and verified. Based on more κ data from horizontal components of all recordings, we re-fitted the data points linearly and obtained a new linear form.

The new figure 3(c) and (d) we plot are given as the following, which shows the positive slope.

We hope the revisions would satisfactory address the reviewers’ concerns.

2.In the Figure 3, the kappa values calculated from records are the same, but the linear fitted lines are different, why?

Response: Thank you for pointing this out.

By checking our drawing data and drawing programs in Matlab, we find that the linear fitted lines and negative slope in original Figure 3(c) and (d) are false. The figure 3 we presented is the original research picture.

We are so sorry that I forget to provide new picture in the manuscript.

The data of κ value and program are checked and verified. Based on more κ value from horizontal components of all recordings, we re-fitted the data points linearly and obtained a linear form in research, but we did not provide these.

The new figure 3 we plot are given as the following, and we provide these in the second round revision.

In the new figure 3 ,the kappa value are re-calculated and the line are re-plot. The new linear fitted lines are given.

3.The authors claim that they replaced “improved method” with “improved model”, but I believe that the improved model is also inappropriate. Why does the word “improved” have to be used when the author has not made any improvements to the method or parameter model? Some improvements to the EXSIM method have been made in the following papers, these may be helpful to you.

(1) Slip-correlated high-frequency scaling factor for stochastic finite-fault modeling of ground motion. Bulletin of the Seismological Society of America, 2022, 112(3):1472-1482.

(2) Simulation of earthquake ground motion via stochastic finite-fault modeling considering the effect of rupture velocity. Stochastic Environmental Research and Risk Assessment, 2023, 37:2225-2241.

(3) An updated stochastic finite fault modeling: Application to the Mw 6.0 earthquake in Jiashi, China. Soil Dynamics and Earthquake Engineering, 2022, 162:107450..

Response: Thanks for the valuable comment. We have read papers the reviewer suggested in the comments.

We fully agree with the reviewers’ assessment.

According to the reviewers’ advice, the expressions such as “improved method” or “improved model” have been deleted or changed in full paper. On the basis of reading and revising the entire manuscript, the expressions such as “improved method” or “improved model” have been revised.

Please refer to the revised manuscript for details on multiple revisions

4.The most serious problem is the kappa estimation. In the manuscript, equation (6) and equation defined in line 238 were mentioned. Which one was used? In addition, equation defined in line

---

## [Decision Letter · Decision Letter 2]

12 Nov 2025

Parameters estimation of stochastic finite fault ground motion simulation method and its application in North China

PONE-D-25-27507R2

Dear Dr. Jia,

We’re pleased to inform you that your manuscript has been judged scientifically suitable for publication and will be formally accepted for publication once it meets all outstanding technical requirements.

Kind regards,

Benedetto Schiavo, Ph.D.

Academic Editor

PLOS ONE

Reviewers' comments:

Reviewer's Responses to Questions

**Comments to the Author**

Reviewer #1: All comments have been addressed

Reviewer #2: All comments have been addressed

2. Is the manuscript technically sound, and do the data support the conclusions?

Reviewer #1: Yes

Reviewer #2: Yes

3. Has the statistical analysis been performed appropriately and rigorously?

Reviewer #1: Yes

Reviewer #2: Yes

4. Have the authors made all data underlying the findings in their manuscript fully available?

Reviewer #1: Yes

Reviewer #2: Yes

5. Is the manuscript presented in an intelligible fashion and written in standard English?

Reviewer #1: Yes

Reviewer #2: Yes

Reviewer #1: (No Response)

Reviewer #2: The authors have addressed all the comments, and the manuscript can be accepted for publication.

The author should have a correct understanding of the calculation process of kappa. kappa = -slope/(pai*loge) and kappa = slope/(pai*loge), kappa = slope/pai, these equations mentioned above are all correct�it mainly depends on how the slope is calculated.

**Do you want your identity to be public for this peer review?** For information about this choice, including consent withdrawal, please see our Privacy Policy

Reviewer #1: No

Reviewer #2: No

---

## [Editor Report · Acceptance letter]

PONE-D-25-27507R2

PLOS ONE

Dear Dr. Jia,

I'm pleased to inform you that your manuscript has been deemed suitable for publication in PLOS ONE. Congratulations! Your manuscript is now being handed over to our production team.

Kind regards,

on behalf of

Dr. Benedetto Schiavo

Academic Editor

PLOS ONE